# Stability Oracle: a structure-based graph-transformer framework for identifying stabilizing mutations

Daniel J. Diaz [1,2,3] ✉, Chengyue Gong[1], Jeffrey Ouyang-Zhang[1], James M. Loy [2,4], Jordan Wells [5], David Yang[4], Andrew D. Ellington [4], Alexandros G. Dimakis[6] & Adam R. Klivans[1]

Engineering stabilized proteins is a fundamental challenge in the development of industrial and pharmaceutical biotechnologies. We present Stability Oracle: a structure-based graph-transformer framework that achieves SOTA performance on accurately identifying thermodynamically stabilizing mutations. Our framework introduces several innovations to overcome well-known challenges in data scarcity and bias, generalization, and computation time, such as: Thermodynamic Permutations for data augmentation, structural amino acid embeddings to model a mutation with a single structure, a protein structure-specific attention-bias mechanism that makes transformers a viable alternative to graph neural networks. We provide training/test splits that mitigate data leakage and ensure proper model evaluation. Furthermore, to examine our data engineering contributions, we fine-tune ESM2 representations (Prostata-IFML) and achieve SOTA for sequence-based models. Notably, Stability Oracle outperforms Prostata-IFML even though it was pretrained on 2000X less proteins and has 548X less parameters. Our framework establishes a path for fine-tuning structure-based transformers to virtually any phenotype, a necessary task for accelerating the development of protein-based biotechnologies.

The ability to predict and understand the change in protein thermodynamic stability ($\Delta\Delta G$) for an amino acid substitution is a core task for the development of protein-based biotechnology, such as industrial biocatalysts[1–3] and pharmaceutical biologics[4–7]. Proteins with enhanced thermodynamic stability are less prone to unfolding and aggregation and are more engineerable[8,9]; stabilizing the scaffold of a protein enables downstream exploration of potentially destabilizing mutations that may improve a target function[8]. Thermodynamic stability is measured by the change in Gibbs free energy ($\Delta G$) between the native and unfolded state and reflects the underlying integrity of the global structure. Engineering proteins is a very laborious process and bottlenecks the development of protein-based biotechnologies[10,11], making the development of computational methods that can accurately predict $\Delta G$ of a point mutation, and in turn identify stabilizing mutations, a highly active research area[12–16].

Deep learning is currently revolutionizing many physical and biological disciplines[17–21], with AlphaFoldV2 leading the way as the scientific breakthrough of the year in 2021[22,23] and sparking an entire wave of deep learning structure prediction tools[24,25]. Although a variety of sequence-based[26–29] and structure-based[30–32] deep learning

[1]UT Austin, Department of Computer Science, Austin, TX 78712, USA. [2]Intelligent Proteins, LLC, Austin, TX 78712, USA. [3]UT Austin, Department of Chemistry, Austin, TX 78712, USA. [4]UT Austin, Department of Molecular Biosciences, Austin, TX 78712, USA. [5]UT Austin, McKetta Department of Chemical Engineering, Austin, TX 78712, USA. [6]UT Austin, Chandra Family Department of Electrical and Computer Engineering, Austin, TX 78712, USA. ✉e-mail: dannyjdiaz305@gmail.com

frameworks for stability prediction have been reported, the lack of data and machine learning engineering issues have prevented deep learning algorithms from having a similarly revolutionary impact on protein stability prediction (see Supplementary Section A for details on these current deep learning frameworks).

Systematic analyses[13,15,16,33] of the state-of-the-art (SOTA) computational stability predictors published over the last 15 years highlight data scarcity, variation, bias, leakage, and poor metrics for evaluating model performance as the critical issues hindering meaningful progress (see Supplementary Section B for a summary on the primary data issues). As a result, the community still primarily relies on physics-based methods, such as Rosetta[34] and FoldX[35], or shallow machine learning approaches[36–44]. While current SOTA computational tools often report 75-80% accuracies upon publication, these accuracies primarily reflect their performance on destabilizing mutations, which make up a majority of the test set and their identification is key for identifying pathogenic variants[13,14,33]. However, when evaluated for predicting stabilizing mutations on the datasets by third parties, about 20% of predictions are actually stabilizing[13,14,33]. Although stabilizing mutations are naturally less common[45], poor generalization to stabilizing mutations has been demonstrated to be a result of pervasive data leakage between train-test splits and severe class imbalance (destabilizing mutations make up >70%) in the current training sets used by the computational stability prediction community and needs to be addressed for machine learning-guided protein engineering[13,14,16,33]. Finally, current computational tools are evaluated with Pearson correlation and RMSE as the primary metrics. Due to the significant imbalance between stabilizing (<30%) and destabilizing mutations in training and test sets and the innate variations associated with measuring $\Delta\Delta G$[15], improvements in these metrics do not necessarily translate into improvements for identifying stabilizing mutations[13,33]. Thus, metrics such as precision, recall, area under receiver operating characteristic (AUROC), and Matthew's correlation coefficient (MCC) should also be considered when developing a model for protein engineering applications[13,14,33] (See Supplementary Section B for a summary of metric issues).

In this work, we address these key issues and develop a robust stability prediction framework for engineering proteins. We present Stability Oracle: a structure-based deep learning framework that makes use of several innovations in data and machine learning engineering specific to stability prediction. Stability Oracle uses a graph-transformer architecture that treats atoms as tokens and utilizes their pairwise distances to inject a structural inductive bias into the attention mechanism. The input to Stability Oracle consists of the local chemistry surrounding a residue with the residue deleted (the masked microenvironment) and two amino acid embeddings to represent a specific point mutation. This design decision enables Stability Oracle to generate all 380 possible point mutations from a single microenvironment, circumventing the need for computationally generated mutant structures and making deep mutational scanning inference computationally inexpensive. To improve the generalization capability of Stability Oracle, we introduce a data augmentation technique—thermodynamic permutations (TP). Thermodynamic permutations, similar to thermodynamic reversibility (TR)[46], are based on the state-function property of Gibbs free energy[47]. For a specific position in a protein, TP expands $n$ empirical $\Delta\Delta G$ measurements into $n(n-1)$ thermodynamically valid measurements, thus increasing the dataset by up to an order of magnitude based on the number of residues that have multiple amino acids experimentally characterized (See Supplementary Fig. 2). Compared to TR, TP enlarges the mutation types sampled across microenvironments in a training or test set, attenuating mutation type bias produced by alanine scanning experiments. Furthermore, unlike TR, it generates a balanced set of stabilizing and destabilizing mutations

to non-wild-type amino acids. This allows us to better assess generalization for protein engineering, as the goal is to mutate away from the wild type. We curate three datasets (C2878, cDNA117K, T2837) to address data leakage issues. MMseqs2[48] was used to generate these three datasets and ensure the maximum overlap between proteins in the training and test set is below 30% sequence similarity, a threshold within the "twilight zone" where 95% of protein pairs will have different structural folds[49]. Concat 2878 (C2878) and Test 2837 (T2837) datasets are new training and test splits from previously established training and test sets, respectively. The third (cDNA117K) is a curated subset of the natural domains from cDNA display proteolysis dataset #1[50], a dataset of ~850 K thermodynamic folding stability measurements across 354 natural and 188 de novo miniprotein domains (40-72 amino acids). This latter dataset is especially interesting because of its size and because it relies upon proteolytic stability as a surrogate for thermodynamic stability. Finally, we generate a sequence-based counterpart for Stability Oracle by fine-tuning ESM2[25] on our curated training and test sets using the Prostata framework[26] and train Prostata-IFML. With Prostata-IFML, we conduct head-to-head comparisons to demonstrate the advantage of structure over sequence-based methods. Overall we show that Stability Oracle and Prostata-IFML are state-of-the-art structure and sequence-based frameworks for computational stability prediction, respectively.

## Results
### Designing a graph-transformer framework for structure-based protein engineering

In prior work, we have experimentally shown that representations learned via self-supervised deep learning models on masked microenvironments (MutCompute[51,52] and MutComputeX[53]) can identify amino acid residues incongruent with their surrounding chemistry. These models can be used to "zero-shot" predict gain-of-function point mutations[51–55], including in protein active sites of computational structures[53]. Self-supervised models, however, generate mutational designs that are not biased towards a particular phenotype and do not update predictions based on experimental data[19].

The MutCompute framework uses a voxelized molecular representation[51,53]. For protein structures, voxelization is a suboptimal molecular representation, as most voxels consist of empty space and rotational invariance is not encoded. Furthermore, the MutCompute frameworks use convolution-based architectures, which lag behind modern attention-based architectures in terms of representation learning and predictive power.

To develop a more powerful and generalizable framework for downstream tasks, we first built MutComputeXGT, a graph-transformer version of MutComputeX (Fig. 1a). Each atom is represented as a node with atomic elements, partial charges[56], and SASA[57] values as features with pairwise atomic distances labeling edges. Our graph-transformer architecture converts the pairwise distances into continuous and categorical attention biases to provide a structure-based inductive bias for the attention mechanism. To generate likelihoods for the masked amino acid, we average the final-layer hidden representations of all atomic tokens within 8Å of the masked $C_\alpha$. The design decision to narrow the pooling to atoms in the first contact shell of the masked amino acid is based on insights from systematic variation of the microenvironment volume when training self-supervised 3DCNNs[58]. With a similar number of parameters and the same train-test splits, MutComputeXGT demonstrates superior representation learning capacity than MutComputeX, achieving wild-type accuracy of 92.98% ±0.26% compared to ~85%[53].

The Stability Oracle architecture makes use of both the feature-extractor and the classification head of MutComputeXGT for supervised fine-tuning (Fig. 1b). Previous structure-based stability predictors[30–32,34,59,60] require two structures—either experimental or

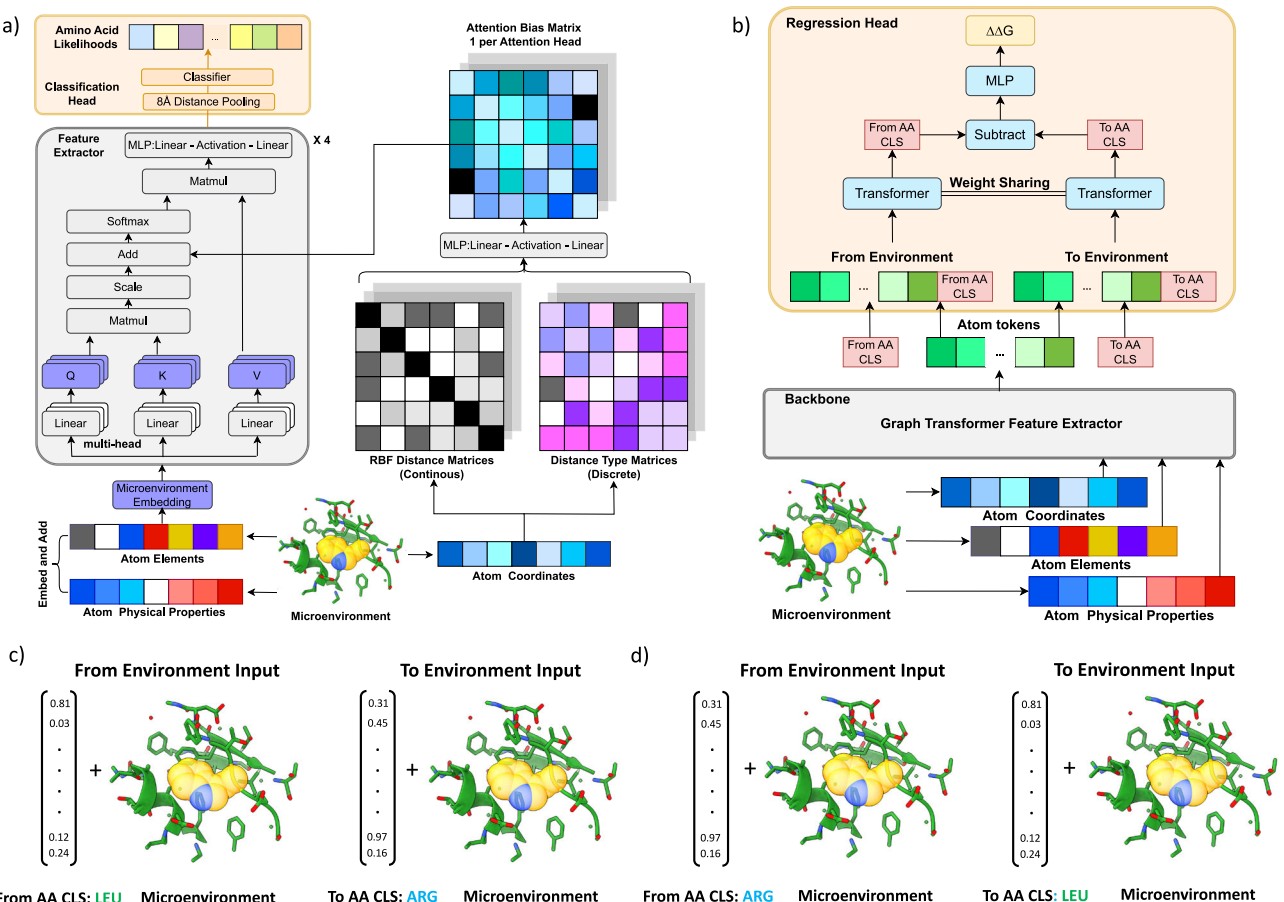

**Fig. 1 | Overview of the Stability Oracle Framework. a** Self-supervised pre-training graph-transformer architecture (MutComputeXGT). **b** Fine-tuning of the pre-trained graph-transformer backbone for stability regression (Stability Oracle). In the regression head, we represent a mutation with "FromAA" and "ToAA" CLS tokens, which are the structural amino acid embeddings for the corresponding amino acids. **c**, **d** demonstrates how Stability Oracle combines structural amino acid embeddings and one masked microenvironment to generate thermodynamic permutations (TP) augmentation mutation inputs. Here, ΔΔG measurements at PDB:5UCE W43 (yellow transparent spheres) for mutations to both LEU and ARG enable the generation of the TP mutations **c** from LEU to ARG and **d** from ARG to LEU by simply swapping the order of the structural amino acid embeddings provided to the regression head. A diagram further describing TP is provided in Supplementary Fig. 2.

computational—to explicitly model the wild type and mutant amino acids. This second (mutant) structure is typically obtained using computational techniques such as AlphaFold or Rosetta. The drawbacks of this approach are (1) computational methods become expensive at inference time (as we describe below) and (2) it is difficult to evaluate the quality of computationally derived mutant structures. In contrast, Stability Oracle does not rely on a second structure. More specifically, structural features from the local chemistry surrounding a particular residue (the masked microenvironment) are extracted from a single initial structure, and a mutation is represented as a pair of "from" and "to" amino acid embedding vectors. To model the ΔΔG of a specific mutation, the microenvironment of the initial structure is used to contextualize the "from" and "to" amino acid embeddings in the regression head (as illustrated in Fig. 1b). This architectural design allows the framework to implicitly learn how "from" and "to" amino acids interact with the local chemistry, rather than relying on a computational structure prediction tool to provide chemical interactions. For a typical 300 amino acid protein, prior work would generate 5700 computational mutant structures (from Rosetta[34] or AlphaFold[22]) in order to predict the ΔΔG of every possible single-point mutation during inference. Stability Oracle, on the other hand, requires only one structure to predict the ΔΔG for all 19 amino acid substitutions at every residue (~50 ms/residue). Runtime performance metrics on proteins of varying lengths are provided in Supplementary Table 1. The "from"

and "to" amino acid embeddings are derived from the weights of the final layer of the MutComputeXGT classifier. This design decision is based on the insight that the weights of these 20 neurons represent the similarity of a microenvironment's features to each of the 20 amino acids prior to being normalized into a likelihood distribution. Thus, they are structure-based contextualized embeddings of the 20 amino acids self-supervised pre-trained from a 50% sequence similarity representation of the Protein Data Bank (PDB)[61], we will refer to them as Structural Amino Acid Embeddings.

We highlight several design decisions of the regression head used in Stability Oracle. Of note is the use of a Siamese[62] attention architecture that treats the mutation embeddings as two classification (CLS) tokens (Fig. 1b). CLS tokens are commonly used in the natural language processing (NLP) community to capture the global context for downstream tasks[63]. Since atoms and amino acids are chemical entities at different scales, we designed the regression head so that a particular microenvironment contextualizes the "From" and "To" amino acid-level CLS tokens. Once contextualized for a given microenvironment, the two amino acid CLS tokens are then subtracted from each other to produce a mutation-hidden representation, which is then decoded into a ΔΔG prediction. This design enforces the state-function of the property of Gibbs free energy[64], providing the proper inductive bias for Thermodynamic Reversibility and "self-mutations" (where ΔΔG = 0 kcal/mol).

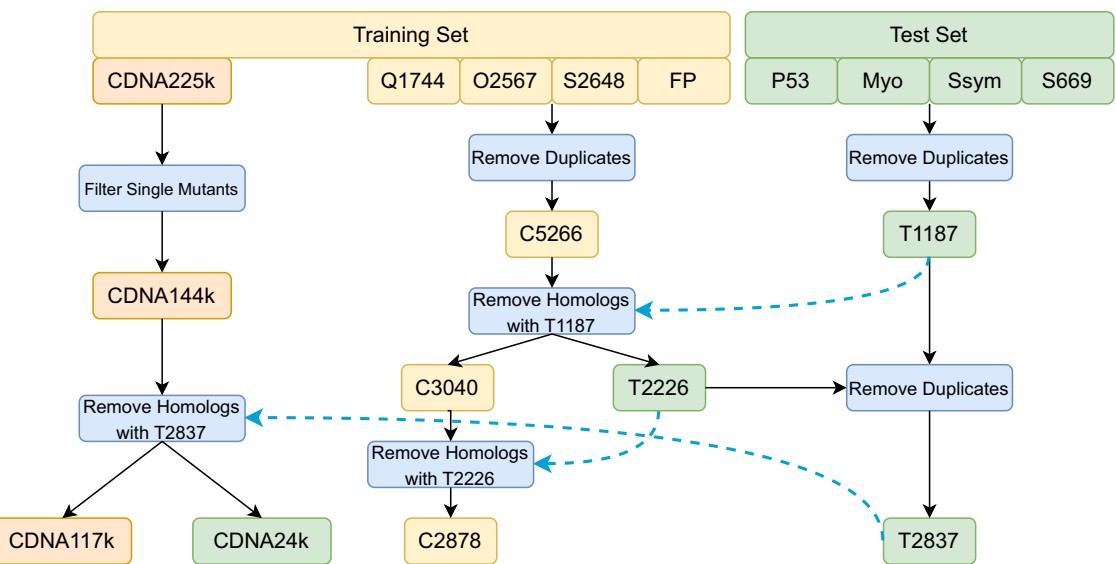

**Fig. 2 | Training and test set generation pipeline.** Homologous proteins were identified using MMSeqs2[48] with a sequence similarity threshold of 30%. Q1744, O2567, and FP (FireProtDB[83]) represent different datasets. See detailed explanations in the main context.

## Training Stability Oracle to generalize across protein chemistry

The Stability Oracle framework was designed to generalize across all 380 mutation types at all positions within a protein structure. The development of such a model has historically been limited by data scarcity, bias, and leakage. To address these issues, we curated training and test datasets and developed a data augmentation technique called thermodynamic permutations (TP).

It is well-known that a major issue with prior works is the inclusion of similar proteins between the training and test set ("data leakage")[42], resulting in poor evaluation of generalization[13,14,33,42]. It has been demonstrated that train-test splits at the mutation, residue, or protein level result in overfitting to the validation set, and strict sequence clustering is required to ensure proper evaluation of generalization[13,14,33,42]. Thus, we created new train-test splits based on a 30% sequence similarity threshold computed by MMSeqs2[48]. First, we built the T2837 test set, which we then used to remove any homologous proteins from the remaining experimental data to produce the C2878 training set. The same procedure was used to construct the cDNA117K training set from the single mutant subset that had experimental structures available of the recently published cDNA-display proteolysis Dataset #1[50] (Fig. 2).

Even with the T2837 expanded test set, we are still unable to assess generalization performance on 14% of the 380 mutation types since they are not represented in T2837. Additionally, T2837 is heavily biased with mutations to alanine (Fig. 3a, bottom row), further hindering our ability to evaluate the generalization of our model. The community has traditionally relied on the data augmentation technique thermodynamic reversibility (TR)[46] to generate datasets with expanded mutation type coverage (Supplementary Fig. 2). However, ~3% of mutation types in C2878 + TR and T2837 + TR still lack data (see Supplementary Fig. 3). More importantly, a major drawback of TR augmentation is that all stabilizing mutations it generates are to the wild-type amino acid, as shown in Fig. 3b. These mutations give no predictive power with respect to identifying non-wild-type stabilizing mutations, which is the main goal of thermodynamic stability prediction in the context of protein engineering. To improve the predictive power of deep learning frameworks for stabilizing proteins, additional data for stabilizing mutations *not* "to" the wild-type amino acid is required.

To address these issues and improve Stability Oracle's ability to generalize, we introduce thermodynamic permutations (TP), a data augmentation technique. TP is based on the state-function property of

Gibbs free energy, enabling the generation of thermodynamically valid point mutations at residues where multiple amino acids have been experimentally characterized. With TP, we can generate an additional 2.02M, 18.9K, and 7.7K point mutations that sample all 380 mutation types for cDNA117K, C2878, and T2837, respectively. Additionally, TP mitigates several sampling biases in all 3 datasets (Fig. 3a, middle column). First, it provides mutation data for the 13.2% and 14.5% mutation types absent in C2878 and T2837. TP generated data for C2878 and T2837 samples of the 380 mutation types, providing the first training and test sets with experimental $\Delta\Delta G$ measurements for all mutation types (the cDNA display proteolysis dataset does not directly measure $\Delta\Delta G$ but instead derives $\Delta\Delta G$ values from the next-generation sequencing data of multiplexed proteolytic experiments).

Figure 3a illustrates the improvement in sampling bias as a softening of red (oversampled) and blue (undersampled) toward white (balanced sampling). In the C2878 and T2837 datasets, this is most apparent for the "to" alanine bias. In cDNA117K, there is an oversampling bias of mutations "from" alanine, glutamate, leucine, and lysine and an undersampling bias for mutations "from" cysteine, histidine, methionine, and tryptophan. TP completely balances the cDNA117K mutation type distribution with each mutation type making up approximately 0.26% of the dataset (100%/380), depicted in Fig. 3a middle column, middle row as uniformly white. Thus, TP augmentation of cDNA117K provides the first large-scale $\Delta\Delta G$ dataset (>1M) that evenly samples all 380 mutation types across 100 protein domains. In contrast to TR, TP does not include stabilizing mutations to the wild-type amino acid and yields a balanced distribution (stabilizing vs. destabilizing) of $\Delta\Delta G$ measurements (Fig. 3b).

To develop Stability Oracle framework, we compared training on cDNA117K and/or C2878, with and without TP augmentation, using Structural Amino Acid Embeddings vs. one-hot encodings, and evaluated the performance on all test sets. We observed that Structural Amino Acid Embeddings significantly improve performance compared to the naïve one-hot encoding in Fig. 4a. UMAP visualization of the mutation-hidden representation for T2837 from the Structural Amino Acid Embeddings reveals that the "ToAA" CLS token drives the organization of the latent space and recover known biochemical relationships between the 20 amino acids as illustrated in Fig. 5a. We observe that 1) clustering of hydrophobes (LEU, VAL, ILE, MET), aromatics (PHE, TYR, TRP), and short polars (SER, THR and ASP, ASN) (right panel); 2) isolation of the unique amino acids (GLY,

(a)

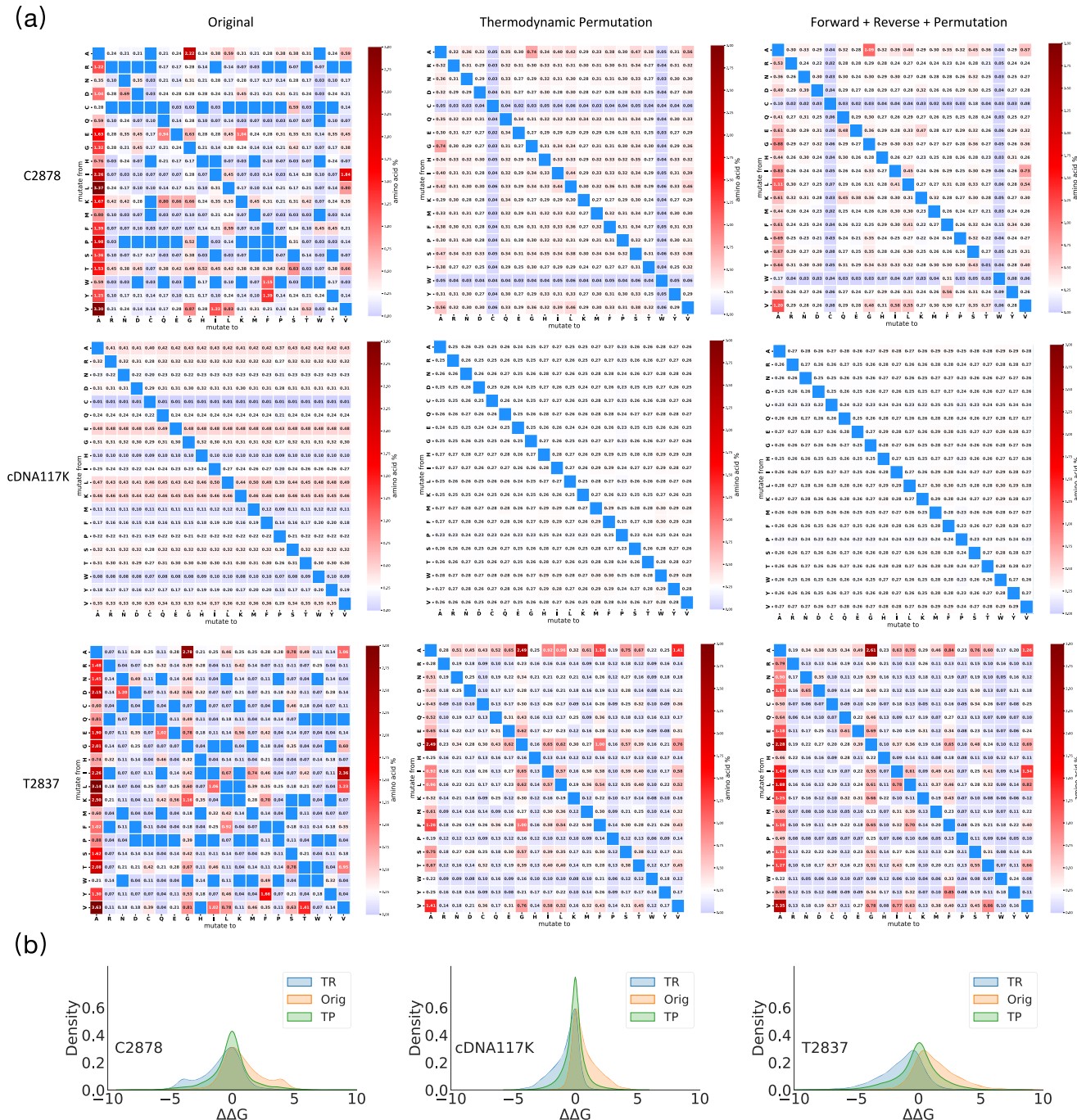

(b)

**Fig. 3 | An overview of the mutation type and ΔΔG distributions for the three proposed datasets and the impact of applying thermodynamic augmentations on these datasets. a** Heatmap representation of the mutation type distribution present in C2878, cDNA117K, T2837. For C2878, the original, TP, and original + TP + TR dataset consist of 2878, 18,912, and 24,668 mutations that sample 86.8%, 100%, and 100% of the mutation types, respectively (first row). For cDNA117K, the original, TP, and original + TP + TR datasets consist of 116,641, 2,018,710, and 2,251,992 mutations, respectively, and each dataset samples 100% of the mutation

types (second row). For T2837, the original, TP, and original + TP + TR datasets consist of 2837, 7720, and 13,394 mutations that sample 85.5%, 100%, and 100% of the mutation types, respectively (third row). **b** Comparing the ΔΔG distribution for the original training and test sets with their TP and TR augmentations. All three experimental datasets are biased towards destabilizing mutations (orig). TR augmentation provides additional data biased towards stabilizing mutations and TP augmentation provides additional data that is evenly distributed between stable and destabilizing mutations.

CYS, PRO) (right panel); 3) the unique situation of mutating away from GLY and adding a chiral side-chain (left panel). For a residue-specific case study of the 380 mutation types, see Supplementary Fig. 7. As for the training sets, training the self-supervised representations on cDNA117K + TP + TR provided the best performance overall on regression and classification metrics across the test sets (shown in Fig. 4b/c). While this might have been expected due to the

sheer size and mutation-type balance compared to C2878 + TP + TR, it is interesting to note that proteolytic stability of single-domain natural proteins is in fact generally an excellent surrogate for thermodynamic stability (as was pointed out in the original publication[50]). From this data, the impact of TP on model generalization was unclear. To further examine how TP-augmented datasets impact generalization, we evaluated predictions at

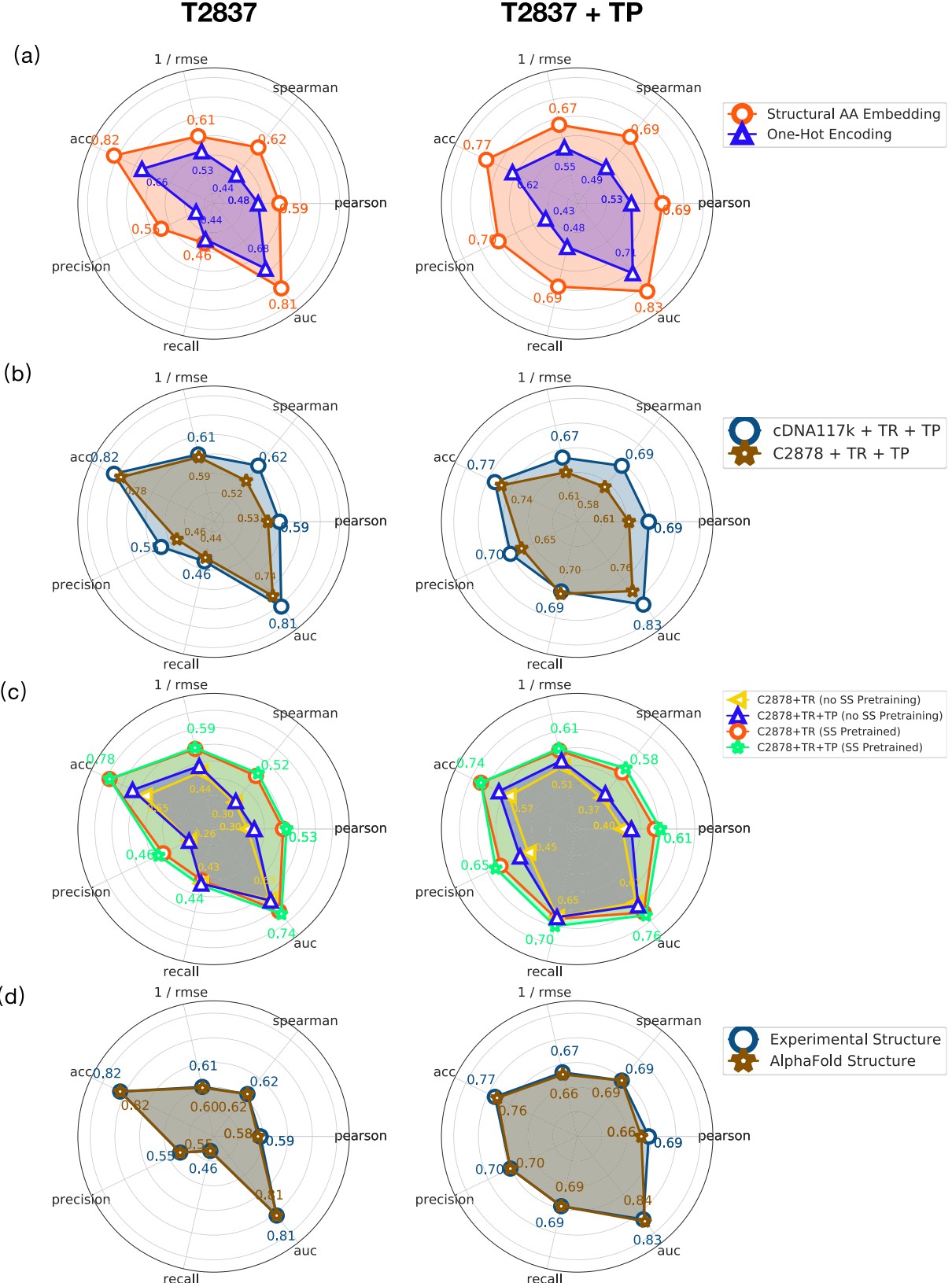

**Fig. 4 | Regression and classification performance of different models on T2837 and T2837 + TP. a** Trained on cDNA117K, we demonstrate the performance with "Structural AA Embedding" and "One-Hot Encoding". **b** Comparison of Stability Oracle's performance when trained on the C2878 and cDNA117K training sets. **c** Comparison of Stability Oracle's performance when trained on C2878 with and without a pre-trained backbone and Structural Amino Acid Embeddings and with and without TP augmentation. **d** Trained on T2837 + TP, Stability Oracle's performance when tested on experimental or AlphaFold structures of T2837. We get *p*-value < 0.001 for all our reported correlation coefficients. Source data are provided as a Source Data file.

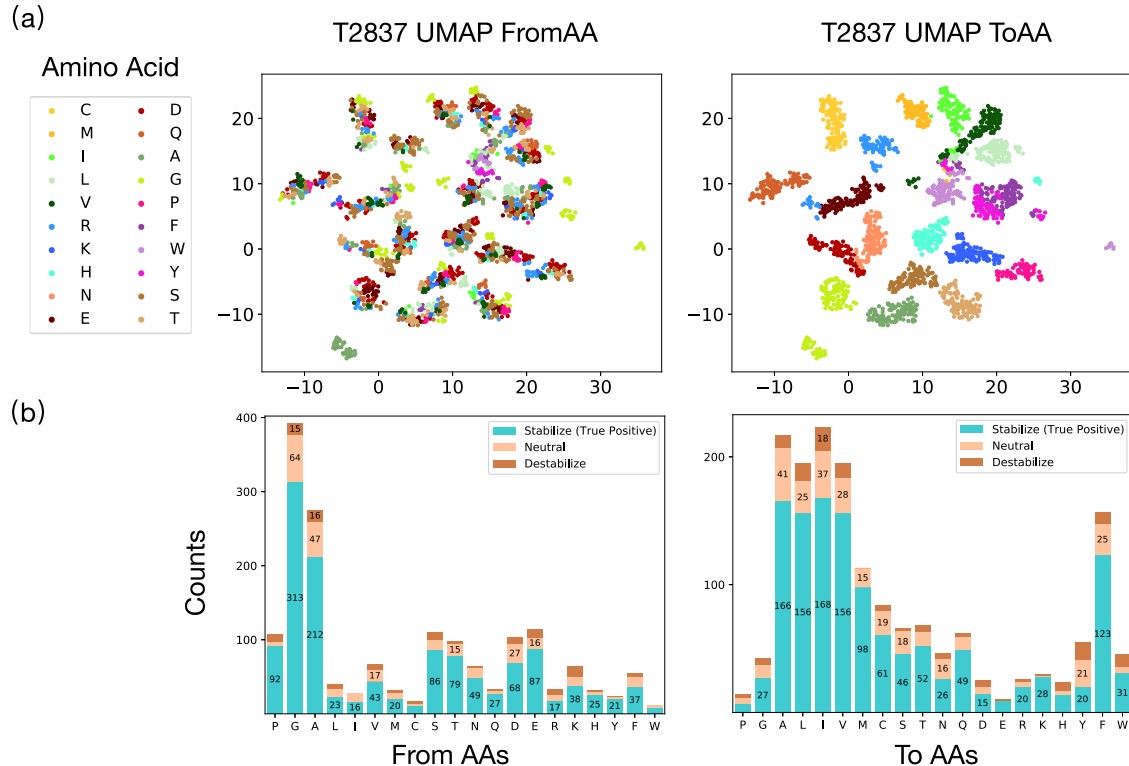

**Fig. 5 | Evaluations of Stability Oracle from the FromAA and ToAA perspective.** **a** UMAP visualization of the 128-dim mutation-hidden representation for T2837. Left and right panels are colored by the "FromAA" and "ToAA" in a mutation, respectively. **b** The experimental distribution of Stability Oracle's stabilizing predictions ($\Delta\Delta G < -0.5$ kcal/mol) on T2837 + TP test set for different "from" and "to" amino acid types. Here, stabilizing, neutral, and destabilizing mutations are defined by $\Delta\Delta G < -0.5$ kcal/mol, $|\Delta\Delta G| \leq 0.5$ kcal/mol, and $\Delta\Delta G > 0.5$ kcal/mol, respectively. Source data are provided as a Source Data file.

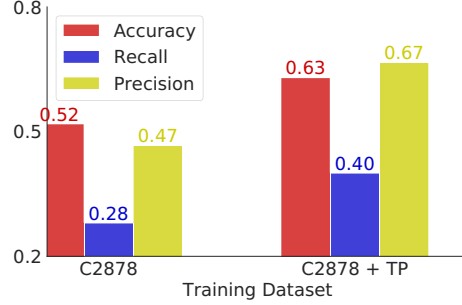

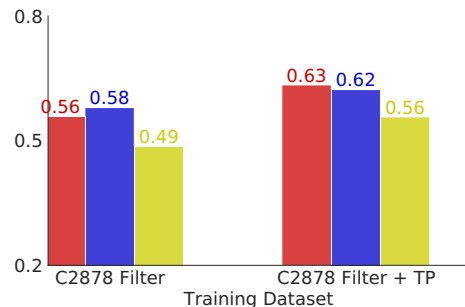

**Fig. 6 | Evaluation of TP on mutation types lacking experimental data in the C2878 training set.** We report the accuracy, recall, and precision results (with 0 kcal/mol being the threshold) on two subsets of T2837 + TP, to demonstrate the effectiveness of permutation. On the left, we test the model on the 12 mutation types lacking experimental data in C2878 + TR (missing mutation types can be found in Supplementary Fig. 3). However, this analysis consists of only 54 mutations within T2837 + TP. On the right, we examine the impact of TP by artificially expanding the missing mutation types from 12 to 68 by removing mutation types from C2878 that had fewer than 8 training instances available. This filtered version of C2878 allowed us to evaluate the performance of TP on 663 mutations within T2837 + TP. Source data are provided as a Source Data file.

mutation types in T2837 + TP that were *absent* from C2878 + TR but *in* C2878 + TP + TR, namely the 12 mutation types with no experimental data (see Supplementary Fig. 3). For these mutation types, TP improves generalization: recall improves from 0.28 to 0.4 and precision improves from 0.47 to 0.67 (Fig. 6). We artificially expanded the mutation types that were missing data to 54 and observed similar, but attenuated improvements to both precision and recall (Fig. 6).

To prevent inflation of Stability Oracle's classification performance, we focus our evaluations on T2837 + TP (10,557 mutations) and exclude all TR mutations, since these mutations are heavily biased with stabilizing mutations to the wild-type amino acid (see Supplementary Fig. 4). Here, Stability Oracle demonstrated a recall of 0.69, a precision of 0.70, and an AUROC of 0.83 (Fig. 4b). Surprisingly, further fine-tuning with C2878 + TP did not improve performance on T2837 or T2837 + TP. Our analysis reveals all proteins in C2878 are homologous (>30% sequence similarity) to at least one protein in cDNA117K and therefore C2878 does not expand the protein space available for training. This observation provides a rationale for the lack of performance improvement observed upon further fine-tuning on C2878. However, C2878 fine-tuning improves performance on the interface subset of T2837: the Pearson correlation improves from 0.30 to 0.35, and the Spearman correlation improves from 0.29 to 0.35 for the interface microenvironments. This improvement is expected since the

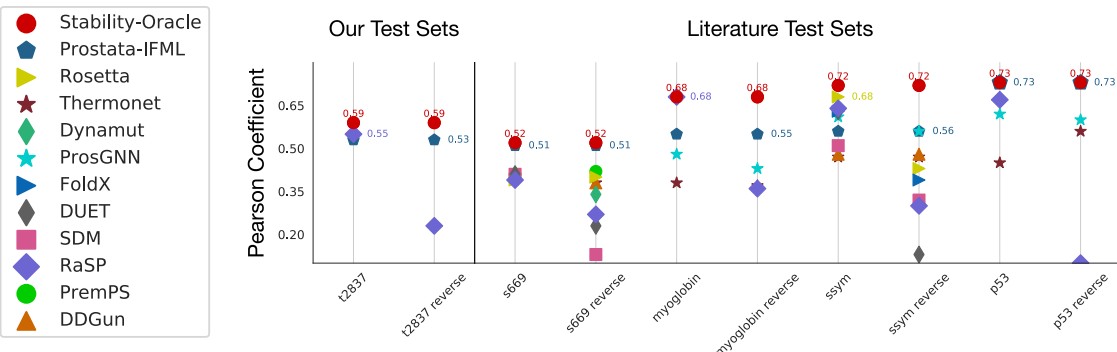

**Fig. 7 | The Pearson correlation coefficient of Stability Oracle and Prostata-IFML across several test sets.** We compare against a handful of computational stability predictors from the community (values obtained from the literature and also provided in Supplementary Table 11[36,38,40,85,86]). Source data are provided as a Source Data file.

cDNA dataset consists of monomeric single-domain structures, lacking interfaces with other proteins, ligands, or nucleotides. However, meaningful improvements are limited due to the scarce amount of protein-protein (127 mutations), protein-ligand (94 mutations), and protein-nucleotide (9 mutations) data in C2878.

Since experimental structures are often unavailable, we examined Stability Oracle's ability to generalize to structures generated by Alphafold2[22] with the WT, "From", and "To" amino acid present. We used ColabFold to generate template-free predicted structures for each protein in T2837[65]. ColabFold failed to fold one protein and two structures were removed due to TM-align[66] having US-scores < 0.5[66,67]. This resulted in the removal of 50 mutations from T2837. When evaluating T2837 and T2837 + TP using the AlphaFold WT structure we observed no changes in classification metrics and slightly lower performance on regression metrics on T2837 + TP (Fig. 4d). Next, we evaluated the impact on the "From" and "To" AlphaFold structures on the T2837 TP-only dataset (7720 mutations, 100% mutation type coverage) and observed a 2–4% drop in classification and regression metrics (Supplementary Table 4). Overall, these results demonstrate the ability of Stability Oracle to generalize to AlphaFold scaffolds when an experimental structure is unavailable.

We conducted several comparisons against the literature. First, we report Pearson correlation coefficients (PCC) (both forward and reverse) on T2837 and all the common test sets. For the common test sets, we compare against several community predictors in Fig. 7 and provide all classification and regression metrics on literature test sets in Supplementary Table 5a. It is worth noting that Stability Oracle outperforms other predictors in the literature even with their documented data leakage issues[14]. To date, the most accurate and exhaustive thermodynamic stability dataset in the literature is the G$\beta$1 dataset[45]. We evaluate Stability Oracle's performance on G$\beta$1[45] and, to the best of our knowledge, achieve SOTA on all 935 mutations (Pearson = 0.75, AUROC = 0.84) and the 835 mutation quantitative subset (Pearson = 0.67, AUROC = 0.81) (full results are provided Supplementary Table 6). Finally, we evaluated Stability Oracle's structural sensitivity with a case study on p53, an issue previously documented for structure-based stability predictors[68]. We evaluated three p53 structures (PDB: 2OCJ, 3Q05, 2AC0) that differed in their protein length (94-312, 94-326, 94-293), resolution (2.05, 2.40, 1.80 Å), and biological assembly (homodimer with no DNA, homo-tetramer complexed with a DNA helix, homotetramer complexed with two DNA helices), respectively, visualized in Supplementary Fig. 1a. This case study demonstrates that Stability Oracle generalizes amid significant structural variations of p53, achieving a Pearson = 0.75 ± 0.02, Spearman = 0.76 ± 0.05, precision = 0.55 ± 0.07, and AUROC = 0.83 ± 0.02 (full results are provided in Supplementary Fig. 1b).

## Evaluating Stability Oracle's ability to identify stabilizing mutations

For computational stability predictors to accelerate protein engineering, it is critical that their predictions correctly identify stabilizing mutations. However, it is well documented that SOTA stability predictors can correctly predict stabilizing mutations with ~20% success rate and most stabilizing predictions are actually experimentally neutral or destabilizing[13,33]. While molecular dynamic-based methods, such as free energy perturbation (FEP), have demonstrated a 50% success rate at identifying stabilizing mutations, their computational demand prevents them from scaling to entire protein applications like computational deep mutational scans (DMS)[33,69]. Thus, there is a strong need for a method that can match the performance of FEP while being computationally inexpensive.

To evaluate Stability Oracle's ability to identify stabilizing mutations, we filtered its predictions on T2837 and T2837 + TP at different $\Delta\Delta G$ thresholds and assessed the distribution of experimental stabilizing ($\Delta\Delta G < -0.5$ kcal/mol), neutral ($|\Delta\Delta G| \leq 0.5$ kcal/mol), and destabilizing ($\Delta\Delta G > 0.5$ kcal/mol) mutations. The 0.5 kcal/mol cutoff was chosen based on the average experimental error[70]. With the $\Delta\Delta G < -0.5$ kcal/mol prediction threshold, 1770 mutations were filtered with an experimental distribution of 74.0% stabilizing, 17.8% neutral, and 8.2% destabilizing and 48.1% of all stabilizing mutations were correctly identified. A systematic analysis of prediction thresholds is provided in Fig. 8 and Supplementary Table 8a. The success rate of predicting stabilizing mutations (74%) appears to surpass what is typically observed with FEP methods (~50%)[33,69] with orders of magnitude less computational cost (Supplementary Table 1). We further examine Stability Oracle's ability to identify stabilizing mutations by amino acid (Fig. 5b). Here, we observe that Stability Oracle is able to correctly predict stabilizing mutations across most amino acids, whether mutating "from" or "to". However, several amino acids lack sufficient "from" or "to" stabilizing predictions to draw meaningful conclusions. This data scarcity is even more apparent when looking at the 380 "from"-"to" pairs (see Supplementary Fig. 5), highlighting how data scarcity still hinders proper model evaluation.

It has been pointed out by the community that experimentally characterized surface stabilizing mutations are biased towards hydrophobic amino acids[33]. An analysis by Broom et al. of the Pro-Therm database[71] indicates that surface stabilizing mutations typically increase side-chain hydrophobicity ($\Delta\Delta G_{solvation}$) with a median change of 0.8 kcal/mol. This hydrophobicity bias is equivalent to an alanine-to-valine mutation on the protein surface[33]. We examined if this underlying hydrophobic bias persisted within our training pipeline by computing the precision and recall of polar and hydrophobic mutations as a function of relative solvent accessibility (RSA) of the wild-type residue. Our precision and recall results across different RSA bins of T2837 and T2837+ TP indicate that the cDNA117K + TP training set

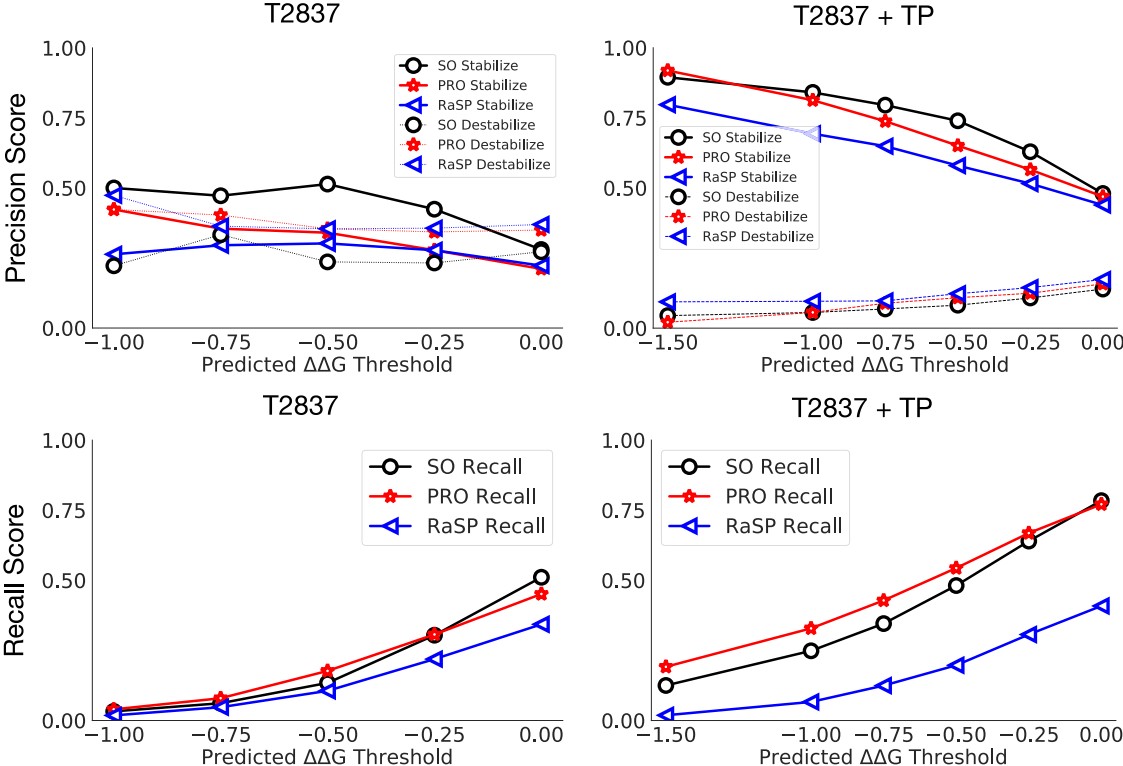

**Fig. 8 | Classification comparisons of Stability Oralce (SO), Prostata-IFML (PRO), and RaSP at different ΔΔG thresholds.** In the first column, we compare T2837 and observe that Stability Oracle had the highest stabilization and lowest destabilization fraction with similar recall. In the second column, we compare T2837 + P and observe that Stability Oracle still has the highest stabilization and lowest destabilization fraction but Prostata-IFML has a better recall. Source data are provided as a Source Data file.

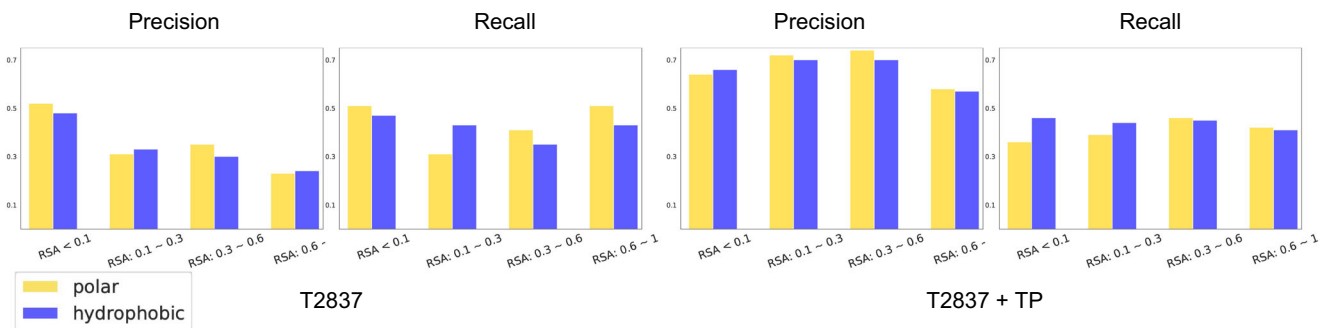

**Fig. 9 | Comparing Stability Oracle's precision and recall between hydrophobic and hydrophilic amino acids as a function of relative solvent accessibility (RSA).** The results demonstrate that there are no biases between polar and hydrophobic residues throughout a protein structure on both T2837 and T2837 + TP: we do a test of significance for the absolute value between polar and hydrophobic data examples on the whole dataset (not per RSA bin due to lack of data) and it is insignificant. Here, polar amino acids consist of S, T, N, Q, D, E, R, K, H, and Y and hydrophobic amino acids consist of L, M, I, V, F, W, A. Source data are provided as a Source Data file.

does not produce models biased towards predicting hydrophobic amino acids on protein surfaces (Fig. 9).

## Comparing sequence and structure fine-tuned stability predictors

Over the last three years, self-supervised protein large language models (pLLMs or "sequence models") have had a tremendous impact on the protein community[25,72–79].

Understanding sequence vs. structure-based prediction models continues to be an active area of research in protein engineering and design[19,80]. We evaluated Stability Oracle against two computational stability deep learning frameworks: Prostata and RaSP. Prostata[26] is a sequence-based framework that fine-tunes ESM2 embeddings

ensembling five distinct regression heads (a SOTA protein language model)[25] on common training and test sets. However, the Prostata was trained with homologous proteins (a sequence similarity cutoff of 75%) with respect to SSym and S669, resulting in inflated performance on T2837 and its subset test sets (breakdown of the performance and data leakage are provided in Supplementary Table 7). In order to address this data leakage and conduct a fair comparison, we fine-tuned ESM2's representation using the same training and test sets as Stability Oracle and only the outer-product regression head architecture. We call our version of ESM2's representations fine-tuned on thermodynamic stability Prostata-IFML. We also compare against the RaSP framework[81]: a structure-based 3DCNN model that follows a similar training pipeline. Briefly, RaSP is pre-trained with self-supervision on 18Å masked

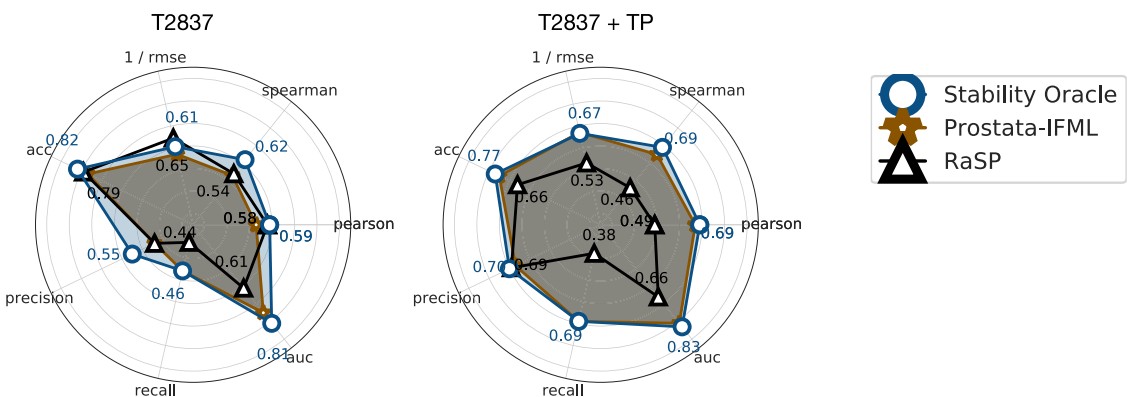

**Fig. 10 | Comparison of Stability Oracle, Prostata-IFML, and RaSP regression and classification performance on T2837 and T2837 + TP.** We refer the readers to Supplementary Section A for detailed results. Source data are provided as a Source Data file.

microenvironments sampled from 2315 structures clustered at a 30% sequence similarity and then fine-tuned on 35 DMS datasets computationally generated by the Rosetta Cartesian-$\Delta\Delta G$ program[81]. In our analysis, we modified the RaSP Colab notebook to generate DMS predictions on every protein in T2837 and every mutation in T2837 + TP.

Stability Oracle outperforms or matches Prostata-IFML on every metric (Fig. 10), even though Stability Oracle has 548 times fewer parameters ( ~658M vs. ~1.2M) and was pre-trained with 2000 times fewer proteins ( ~ 46M vs. ~ 23K) at the same sequence similarity (50%). As for RaSP, Stability Oracle significantly outperforms it on nearly every classification and regression metric on both T2837 and T2837 + TP. The performance is only comparable for Pearson on T2837 and precision on T2837 + TP.

In terms of identifying stabilizing mutations, Stability Oracle also achieves the best performance (Fig. 8). At each prediction threshold, Stability Oracle had the highest proportion of correctly identified stabilizing mutations and the lowest proportion of destabilizing mutations (we exclude −1.5 kcal/mol threshold for T2837 due to data scarcity). It is typical for precision to be inversely proportional to recall, and we observe this tradeoff with regard to Stability Oracle and Prostata-IFML on T2837 + TP, where Prostata-IFML has better recall. We suspect this difference is due to their loss: Stability Oracle uses a huber loss and Prostata uses mean square error. Nonetheless, both Stability Oracle and Prostata-IFML have superior performance on both correctly identifying stabilizing mutations (precision) and identifying more stabilizing mutations (recall) compared to RaSP. A detailed comparison is provided in Supplementary Table 8. In parallel to this work, ThermoMPNN–a deep learning framework that fine-tunes the ProteinMPNN[82] representations also on the megascale cDNA proteolysis dataset–was developed. Using the publicly available checkpoint, we found that Stability Oracle outperforms ThermoMPNN on SSym, S669, myoglobin, and P53 across multiple regression and classification metrics (Supplementary Table 10).

Finally, we compare all three framework's ability to predict self-mutations: the "from" and "to" amino acids are the same and the $\Delta\Delta G$ is 0 kcal/mol. Similar to the forward vs. reverse experiments, which assess the thermodynamic robustness of predictors, self-mutations evaluate generalization to trivial examples that were not present in the training set but are inherent in thermodynamics. For wild-type self-mutations on T2837 Stability Oracle, Prostata-IFML, and RaSP achieve RMSE of 0.0033, 0.0018, and 0.8370 kcal/mol, respectively. This demonstrates that Stability Oracle and Prostata-IFML implicitly learn to capture self-mutations. RaSP, however, is unable to generalize to self-mutations and this drop in performance is also observed for TR augmentation of T2837 (Supplementary Table 9c).

## Discussion

Reliable prediction of stabilizing mutations is critical for the acceleration of protein-based biotechnologies. However, as of March 2023, all computational stability predictors prioritize improvements in the Pearson and RMSE metrics. Neither of these metrics, however, is appropriate for evaluating improvements in identifying stabilizing mutations; several studies explain this in great details[13,14,33]. We report these regression metrics but use classification metrics, such as AUROC, MCC, recall, and precision, to guide our model development. By using a 30% sequence similarity threshold for train-test splits, we expect the performance of Stability Oracle and Prostata-IFML to have superior generalization compared to models trained using traditional train-test splits, which suffer from data leakage. Stability Oracle and Prostata-IFML seem to correctly identify stabilizing mutations that potentially outperform FEP-based methods while being several orders of magnitude faster. However, systematic experimentation that directly compares FEP-based methods is needed for confirmation.

The literature has recently adopted thermodynamic reversibility (TR) to address the imbalance between stabilizing and destabilizing mutations in training and test sets. However, TR biases the training towards stabilizing mutations of wild-type amino acids. For computational models that leverage evolutionary features and protein structures as input, mutations to wild-type amino acids leak information and are of limited use in a protein engineering context where mutating away from the wild type is the primary goal. Our data augmentation technique, thermodynamic permutations (TP), generates mutations with a balanced $\Delta\Delta G$ distribution and does not generate mutations to wild-type amino acids (Supplementary Fig. 3b). This mitigates the above imbalance and expands the number of stabilizing mutations to non-wild-type amino acids in both the training and test sets. In addition, TP reduces bias to wild-type inherent in the self-supervised pre-training step. We speculate that during end-to-end fine-tuning, TP forces the feature-extractor to find additional chemical signatures in the microenvironment, not just those most relevant for wild-type identification. Further, TP also generates $\Delta\Delta G$ measurements for mutation types in microenvironments that would rarely be experimentally characterized. This expands the microenvironment/ mutation type combinations available for training and testing and is highly likely to improve generalization across structural motifs found within a protein scaffold. However, we lack the necessary test data needed to examine this hypothesis. We expect TP to be of great use for the development of frameworks for higher-order mutations, where data scarcity is an even bigger issue.

Of note is the performance of Stability Oracle's much smaller model (in terms of parameters) relative to Prostata-IFML. Stability Oracle's ability to outperform or match Prostata-IFML is evidence that protein structures contain information beyond the amino acid

sequence itself. Self-supervised deep learning models, such as Mut-ComputeX, typically struggle with predicting mutations in the core, with a tendency to re-predict the wild-type residue in the closely packed environment. Stability Oracle is explicitly trained to learn stabilizing substitutions from the microenvironment, providing a structure-based method for predicting core mutations.

In terms of training set size, note that the ~25,000 $\Delta\Delta G$ measurements (C2878 + TP + TR) led to performance comparable to models trained on ~2.2 million proteolytic stability measurements (cDNA117K + TP + TR). While T2837 is still a limited evaluation set, this indicates that both the quality and quantity of data are of great importance for training. Additional experimental techniques directed towards producing larger datasets with detailed thermodynamic information, especially for residues at functional interfaces, may further improve generalization capabilities.

Accurate identification of stabilizing mutations will continue to impact a wide variety of areas, from predicting protein therapeutics and vaccines with greater stabilities and shelf lives to predicting enzymes that can work at higher temperatures for biomanufacturing and environmental bioremediation. While previous algorithms, such as MutCompute, have proven adept at improving the stabilities of proteins[51,53,54], Stability Oracle will likely further improve the hit rate of functional, stable substitutions due to its improved accuracy across a variety of metrics. More importantly, Stability Oracle is attuned to thermodynamic effects, rather than just sterics. This allows for predictions across a wider class of substitutions, including those at protein: protein interfaces, such as antibody: antigen interactions, as well as those at protein: ligand and protein: nucleotide interfaces.

## Methods
### Model architecture
We build a transformer-based neural network for both mask amino acid self-supervised learning and $\Delta\Delta G$ supervised learning. We describe the model architectures for the self-supervised backbone model and for the $\Delta\Delta G$ regression head.

### MutComputeXGT: self-supervised graph transformer for amino acid likelihoods
We first introduce our graph-transformer model for self-supervised tasks, outlining its key components and specifications. We describe the general pictures of our model, and then we list the model inputs and outputs. Finally, we elaborate on some key inductive bias designs in our model. Generally speaking, in our tasks, the input for the model is a local environment for the target amino acid. Given one amino acid, we set the Carbon-$\alpha$ as the center and grab all the atoms within radius $n$. In the context of self-supervised training, we apply masking to each atom within the target amino acid and predict the corresponding amino acid type. This approach leverages graph-based representations and self-supervised learning to capture important structural features and enhance our understanding of protein sequences. Denote $N$ as the number of tokens, the coordinates $\in \mathbb{R}^{N \times 3}$, atom types $\in \mathbb{R}^{N \times 1}$ and physical properties $p \in \mathbb{R}^{N \times P}$ are given to the neural network, we apply an embedding layer to convert categorical atom types into continuous representations $e \in \mathbb{R}^{N \times E}$ where $E = 20$ represents the number of amino acid types and concatenate embeddings together with physical properties $h = \text{Concat}(e, p)$. The coordinates $\in \mathbb{R}^{N \times 3}$ are used to calculate atom-wise Euclidean distance $D \in \mathbb{R}^{N \times N}$, which serves as the attention bias in the attention layers. The concatenated features $h$ then pass through several attention blocks. In each attention block, we have two attention layers and one MLP layer. The MLP layers are the same as the standard attention block, while in each attention layer, after calculating the $KQ^T$, we add additional attention mask and attention-bias terms. The attention bias is calculated based on the distance matrix $D \in \mathbb{R}^{N \times N}$. Given $D$, we create multiple feature vectors and output the attention bias for each head in one attention layer.

Apply $K = 16$ radial basis functions (RBF) with different bandwidths, we get $k \in \mathbb{R}^{N \times N \times K}$. Categorizing the distance into $C = 4$ class, we get $c \in \mathbb{R}^{N \times N \times C}$. Put a linear transformation after concatenating $c$ and $k$, we get bias $\in \mathbb{R}^{N \times N \times \text{Head}}$ where Head denotes the number of heads in one attention layer. Finally, we collect the hidden representations $h$ for all the atoms and select those whose corresponding atom is close to the masked amino acid. Specifically, we set the carbon-$\alpha$ for the masked amino acid as the center and select all the atoms in an 8 Å radius. The features of these atom tokens are average pooled, and passed to the classifier, before being normalized into the 20 amino acid likelihoods. The classifier consists of a Linear-ReLU-Linear block, with 128 and 20 neurons at these linear layers, respectively.

### Stability Oracle: graph-transformer fine-tuned for $\Delta\Delta G$ prediction
The Stability Oracle framework contains two parts, a backbone that encodes the masked local environment and a regression head. For the purpose of this discussion, we primarily focus on the architecture of the regression head, as the pre-trained weights from the self-supervised learning task are loaded into the backbone. We get one masked local environment together with two amino acid types (one is the wild-type and the other is the mutation), and target at predicting the $\Delta G$ changes. We first extract useful atomic features from the backbone model. Given a masked local environment, we pass it through the backbone and get the output $h \in \mathbb{R}^{N \times H}$ where $H$ denotes the hidden dimension. Given the wild-type and mutation amino acid type, we extract the corresponding amino acid embedding $e_{wt} \in \mathbb{R}^{1 \times H}$ and $e_{mut} \in \mathbb{R}^{1 \times H}$ in the final linear layer of the backbone. We then apply Concat($h, e_{wt}$) and Concat($h, e_{mut}$) to get two hidden representations for wild-type local environments and mutated local environment, respectively.

Pass Concat($h, e_{wt}$) and Concat($h, e_{mut}$) through the attention blocks, we extract the $e_{wt}$ and $e_{mut}$ in the final layer, subtract them, apply a linear layer, and output the predicted $\Delta\Delta G$.

### Training configuration
During training, we shuffle the training data and use Huber loss with $\delta = 1$ as the training loss. When training on large-scale datasets, e.g., cDNA117K, we end-to-end tune all the parameters, with a learning rate of $5 \times 10^{-5}$ for regression head parameters and a learning rate of $2 \times 10^{-5}$ for backbone parameters. We apply optimizer AdamW with batch size 960 (batch size 240 with accumulation step 4), weight decay 0.1, optimizer EMA with $\eta = 0.99$, and number of iterations 750. To fine-tune the CDNA-trained model on the small datasets (e.g., C2878), we freeze the backbone parameters and use AdamW with a learning rate $5 \times 10^{-7}$, batch size 1,024, weight decay 0.1, and number of iterations 500. Train from scratch on the small datasets (e.g., C2878), we freeze the backbone parameters and set optimizer AdamW with $5 \times 10^{-5}$, batch size 1,024, weight decay 0.1, and number of iterations 500.

### Pre-training dataset
We pre-train our graph-transformer backbone using the same procedure as MutComputeX[53]. Briefly, this dataset consists of a 90:10 split of 2,569,256 microenvironments sampled from 22,759 protein sequences clustered at 50% sequence similarity and having a structure resolution of at least 3 Å from the RCSB (November 2021).

### cDNA117K training set generation
To curate the cDNA117K dataset[50], we downloaded the file K50_dG_Dataset1_Dataset2.csv from the version 1 deposit at the *Zenodo* repository https://zenodo.org/record/7401275#.Y6st59JBxD_. First, we removed all de novo domains and then filtered all single-point mutations on protein scaffolds that have a wild-type structure pdb id provided in the csv. Finally, using structure sequences, the remaining miniproteins were filtered for 30% sequence overlap with T2837 via MMSeqs2 easy-search command with flags -c 0.3 -s 7.5 -seq-id-mode 1. The closest sequence similarity between cDNA117K and T2837 is 25.4%.

## C2878 training set generation

To curate C2878 dataset[50], we first concatenated Q1744[30], O2567[16], S2648[40], and FireProtDB[83]. We address duplicates between the datasets by taking the $\Delta\Delta G$ measurement with the highest absolute value. This was intended to address the neutral bias and move the $\Delta\Delta G$ predictions from a model away from 0. With duplicates removed, we generate C5266, which is then filtered for 30% sequence overlap with T2837. The structure sequences were used for the sequence similarity filtering. The same MMSeqs2 command was used as cDNA117K.

## T2837 test set generation

To curate T2837 dataset[50], we first concatenated P53[30], Myoglobin[30], SSym[30], and S669[12]. We address duplicates between the datasets by taking the $\Delta\Delta G$ measurement with the highest absolute value. This was intended to address the neutral bias and move the $\Delta\Delta G$ predictions from a model away from 0. With duplicates removed, we generate T1187. The 30% sequence overlap with C5266 was removed to produce T2226. T2226 was then merged and deduplicated in the same manner to produce T2837. The structure sequences were used for the sequence similarity filtering.

## Generation of T2837 AlphaFold structure

We ran ColabFold 2.3 on the full UniProt sequence associated with each PDB ID. For the TP-only "From" and "To" datasets, We modified the wild-type UniProt sequence to reflect each mutation. We used ColabFold's default parameters and chose the folded PDB structure with the highest pLDDT score for each sequence.

## Test set

To fully test different models, we collect and clean related datasets in the literature and make our test set. To get more comparisons, we report the numbers on all the literature datasets here. In summary, we test our model on S-sym, P53, Myoglobin and S669. S-sym, P53, and Myoglobin datasets are proposed in ThermoNet[30] and are widely used by the follow-up works as the testing benchmarks. S669 includes curated data dissimilar at 25% sequence identity to S2648 training data. Additionally, we create several more domains from the original test data. ① Swap the mutation and wild-type amino acids, and this test domain is marked as TR. ② Once we have multiple single-point mutations in the same position, we randomly pick two amino acids as the wild-type and the mutation. We mark this test domain as TP.

## Evaluation metrics

We evaluate regression and classification metrics. First, for a fair comparison, we report Pearson correlation and RMSE for our regressor. Furthermore, in practice, instead of RMSE, we have more interest in the performance of stabilizing mutations. Therefore, we report the precision and recall for the stabilizing mutations and the AUROC values for the binary classification problems.

## Prostata-IFML details

We construct a strong sequence-based baseline from Prostata[26]. This baseline model takes the wild-type and mutant sequences as input. Both sequences are first tokenized and independently passed through the same pre-trained ESM2 backbone. The features at the mutation position are selected from both the wild-type and mutant sequences. An outer product enables interactions between the wild-type and mutant embeddings. The result is flattened and a small decoder predicts a change in thermodynamic stability. This is referred to as the outer-product variant in PROSTATA. For training, we employ the identical cDNA dataset that was employed to train Stability Oracle. Given the relatively shorter length of cDNA protein sequences, we were able to increase the batch size significantly to 64 mutations, as opposed to the original batch size of 1. We trained our model using the Adam optimizer for 3 epochs. During the training, we followed a one-cycle learning rate schedule. In the first epoch, the learning rate was linearly increased from 0 to $6.4 \times 10^{-4}$. The learning rate was gradually annealed to 0 in the subsequent epochs. We fine-tune the same training sets and evaluate our models on the same test sets as above.

## ThermoMPNN details

We use the official codebase on GitHub at https://github.com/Kuhlman-Lab/ThermoMPNN with the official checkpoint at https://github.com/Kuhlman-Lab/ThermoMPNN/blob/main/models/thermoMPNN_default.pt to generate predictions for model comparisons.

## RaSP details

We use the codebase on GitHub at https://github.com/KULL-Centre/_2022_ML-ddG-Blaabjerg to generate predictions for model comparisons.

## Reporting summary

Further information on research design is available in the Nature Portfolio Reporting Summary linked to this article.

## Data availability

All relevant data supporting the key findings of this study are available within the article and its Supplementary Information files. Datasets are available on the *Zenodo* repository[84] at the GitHub repository: https://github.com/danny305/StabilityOracle. Source data are provided in this paper.

## Code availability

Code used to generate figures and the Stability Oracle model weights are available on a Zenodo repository[84] and at the GitHub repository: https://github.com/danny305/StabilityOracle.

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

## Acknowledgements

This work was sponsored by grants from the Defense Threat Reduction Agency (HDTRA1201001), the Welch Foundation (F-1654) and the NSF AI Institute for Foundations of Machine Learning (IFML). We would like to thank AMD for the donation of critical hardware and support resources from its HPC Fund.

## Author contributions

D.J.D., A.K., and A.G.D. conceived the project. D.J.D. designed the data engineering pipeline for pre-training and fine-tuning with support from J.M.L. and J.W. D.J.D. developed the training and test set splits. D.J.D. and C.G. developed the graph-transformer architecture and trained Stability Oracle. J.O.Z. trained and evaluated Prostata-IFML. D.Y. generated the AlphaFold datasets. The manuscript was written primarily by D.J.D. and C.G. with support from J.O.Z., A.K., A.G.D., and A.D.E. D.J.D., C.G. and A.K. supervised all aspects of the study.

## Competing interests

D.J.D. and J.M.L have financial relationships with Intelligent Proteins LLC. C.G., J.Z., J.W., D.Y., A.D.E., A.D., and A.R.K. declare that they have no conflict of interest.
