## [Peer Review File · Nature Communications]

Stability Oracle: A Structure-Based Graph-Transformer Framework for Identifying Stabilizing MutationsReviewer #1 (Remarks to the Author):

In "STABILITY ORACLE: A STRUCTURE-BASED GRAPH-TRANSFORMER FRAMEWORK FOR IDENTIFYING STABILIZING MUTATIONS", the authors assert that their Stability Oracle pipeline surpasses other stabilising mutation predictors, citing: 1) its structure-based graph-transformer design, and 2) a generalised dataset developed from Thermodynamic Permutations (TPs) on a vast protein stability dataset. While the manuscript elaborates on the thermodynamic permutation process, ambiguities remain around the TP concept and the methodology employed.

[General]

The Thermodynamic Permutations (TP) concept as an augmentation for $\Delta\Delta G$ is both novel and intriguing. However, it raises a few questions.

The pipeline seems to focus solely on the microenvironment derived from the wild-type for structural features of all 380 TPs.

How different are those structural features across all 380 TPs? Can you give an example?

Would authors expect better performance if the pipeline can take the structural features from even mutants?

What features would mostly contribute to the performance of the Stability Oracle?

Regarding the concept of TPs, for example, if Y->A results in $\Delta\Delta G_1$ and Y->D yields $\Delta\Delta G_2$, the computed difference, $\Delta\Delta G_2 - \Delta\Delta G_1$, might imply a D->A mutation. Yet, if one were to adhere to the thermodynamic cycle, the sequence would be D->Y->A. This indicates that the true effective mutation can be Y->A rather than D->A. In this sense, is it correct that the application of TP seems restricted to the transformer attention layer, particularly the "from" and "to" segments, given that the structural disparity between wild and mutant forms remains unchanged for both TP and TR?

[Others]

[P2, 39] Rosetta is a physical-based not a knowledge-based method.

[P2, 42-43] What is the criteria of the data leakage?

[P2, 47-49] The author highlighted the challenges of using regression metrics on unbalanced datasets, a characteristic intrinsic to this type of dataset. However, while the MCC addresses imbalance, it doesn't capture the predictive accuracy of regression tasks, specifically the precision in predicting $\Delta\Delta G$ values. Also, based on the results, Pearson's correlation coefficients showed reasonably similar to Spearman's rho, even for non-TP test datasets.

[P2, 53-58] Does the Stability Oracle pipeline produce all 380 potential TP cases from just one wild-type structure? If so, does the platform solely incorporate structural features from the wild-type PDB structures? While this approach might address the scarcity or bias of $\Delta\Delta G$ labels, it may not fully account for variations in structural features.

[P2, 65-66] My comprehension is that TP is introduced when a residue has multiple mutational $\Delta\Delta G$ values. For instance, if there are two mutational $\Delta\Delta G$ values, such as $\Delta\Delta G$ Y->A and $\Delta\Delta G$ Y->K, then we can derive $\Delta\Delta G$ K->A and $\Delta\Delta G$ A->K using TP. However, Alanine Scanning results in a single mutation for each residue (X->A), making it unsuitable for the TP process.

[P2, 69-71] Did the author suggest that data leakage can be effectively managed by limiting sequence similarity between the training and test sets to 30%? In the introduction, the authors highlighted that data issues are prevalent and difficult to address using methods like clustering. Therefore, a comprehensive explanation is warranted. Is there empirical evidence to confirm that the 30% threshold is ideal for preventing data leakage?

[P2, 73-74] How many mutations are there in natural versus de novo proteins?

[P2, 77] I'm unable to determine what "Prostata-IFML" is and how it differs from "Prostata." Could you provide more information or clarification?

[P2, 86-88] Do the authors suggest that self-supervised models produce unbiased models even on uneven datasets?

[P3, 109] AlphaFold is better suited for modelling general proteins rather than specific point mutants. ("Can AlphaFold2 predict the impact of missense mutations on structure?")

[P4, Figure 2] No explanation for FP.

[P5, 161-163] Closing parenthesis is missing

[P5, 174] Figure 4 should come earlier than 5

[P5, 180] Is the C2878 + TR correct ? or should it be C878 + TP?

[P5, 181-182] In the test data (T2837 + TP), there are 12 mutation types (e.g., Ala to Tyr) that are not present in the C2878 training dataset, nor in the combined C2878 + TP dataset. Is that correct? Can authors describe what are the 12 mutation types?

[P5, 182-183] The authors described "...observed similar results", but there is a notable difference in performance between using (0.28 -> 0.40) and not using (0.58 -> 0.62) a filter for the training dataset. Additionally, why wasn't the MCC metric included, especially considering the authors' primary assertion that the three metrics ("Accuracy," "Recall," and "Precision") are not reliable?

[P5, 188] It's somewhat unclear; I'd suggest sticking with a 30% sequence homology for clarity.

[P7, 198-200] Could the observed behaviour be because the Stability Oracle does not rely on structural features as much? What do the authors think about this?

[P8, Figure 6] Could the authors consider modifying the symbol representations for clearer visual presentation?

[P8, 220] The metrics used for the Y-axis in Figure 10 and those in Table 5a are unclear.

[P8, 220-222] Is the Stability Oracle's success rate for stabilising mutations 48.1%? If that's the case, it doesn't outperform the "FEP methods" which are around 50%. Moreover, if "success rate" is synonymous with accuracy, wouldn't using the MCC provide a more precise measure? Also, it would be more informative if the Y-axis represented the TP rate instead of Counts, facilitating a smoother comparison of performance across different amino acids (AAs).

[P9, Figure 8] The authors referred to the performance difference as insignificant, but how does this apply to the case of T2837, especially for Recall RSA within the range of 0.1 to 0.3?

[P27, 651] Same as earlier, Rosetta is a physics-based approach.

[P27, 660-662] Same as earlier, AlphaFold2 may not be optimal for generating single-point mutants. Approaches based on homology modelling might be more effective. If the authors recommend using AlphaFold2/Rosetta over MODELLER or PyMOL for mutagenesis, there should be justification highlighting the superior performance of AlphaFold2/Rosetta.

[P28, 697] Could you clarify what "game-ability" refers to in this context?

[P28, 699] Could you clarify what "overly simplified model" is in this context?

[P28, 698-700] The authors utilised the "Spearman" and "MCC" metrics, but they have not provided a clear comparison to demonstrate their superiority over the original metrics.

Reviewer #2 (Remarks to the Author):

In "STABILITY ORACLE: A STRUCTURE-BASED GRAPH-TRANSFORMER FRAMEWORK FOR IDENTIFYING STABILIZING MUTATIONS", Diaz et al. Present a new method for the prediction of stability changes in proteins upon mutation. This is a longstanding field with a plethora of known methods, and while the manuscript presents some interesting new ideas, there are also a number of issues that need to be addressed.

There's a number of issues in relation to the existing literature that makes one wonder if the authors misunderstand the general goal of stability predictions in this field. The most common effect of a mutation on a protein is loss of stability (not gain), and this is a very valuable insight in many areas from biotechnology to interpretation of pathogenic variants. Thus, the perceived discrepancy in lines 40-41 is largely due to the fact that most variants are destabilising, and that these are often predicted correctly. This also explains the imbalance in test/train sets. These are all well-known and long-standing issues - they are properties of (naturally evolved) proteins.

Along similar lines, the authors state on p3 that "previous structure-based prediction programs need 2 input structures, mutant and wild-type" - this is either wrong or at least misleading. The cited papers are from 2019-2020, while structure-based stability prediction has been around since the early 2000s at least (Guerois et al. 2002; Kortemme et al. 2004; Kellogg et al. 2011). All these tools use one input structure, the other is generated - and the structural accuracy of these mutant structures (or lack thereof) is a longstanding issue and not taken lightly.

This leads into the second major issue: in suppl data A.3 - data augmentation the authors illustrate how they combine mutations in some way to generate additional input/training data, called "thermodynamic permutation". While in principle this is possible, it should be described in specific examples how this is carried out. The main questions though are - how do the authors have access to starting structures for these permutations? A challenge in symmetrising classic ddG

approaches is that the very starting structures for the Mut -> wt direction are missing. It would not be fair to use structures generated by the tool itself, reverting mutations are far easier to predict. There are sets with both wt and mutant structures, but they are far smaller, definitely too small for deep learning. Even AlphaMissense does not predict structures (just scores), which may serve as another indication that this is a hard problem. Further, even if structures are available, the local environment of these mutations is going to be extremely similar, hence the actual increase in information will not be massive. It would mainly serve to achieve symmetry.

The currently best-performing tool is ThermoMPNN <https://doi.org/10.1101/2023.07.27.550881> - the authors need to report performance compared to that.

RaSP predicts ddGs in the (-1,7) range and is therefore not really made for stabilising mutations, the focus is on destabilising ones. Hence its lower performance in the benchmarks is not surprising.

Further, the most accurate ddG dataset is that on proteinG (Nisthal et al. 2019), which the authors should also benchmark on

line 69: for "new test sets", it should be very clear what were the original data that went into this, how many sequences there were, how many were removed. Also the new datasets should be provided - the github repository is empty!

A.6 predicted stabilising mutations - many of these are from Gly to hydrophobic residues, based on experience this can be a massive source of error, have the authors tried to validate this in any way?

Fig. A10: this shows a big cloud, please provide an estimated error

Regarding evaluation using Pearson, this is another well-known issue. If the authors are convinced that Spearman is more suitable, they should calculate and report Spearman correlation coefficients and not report Pearson yet over again.

Category analysis is a somewhat more complicated issue. While it is a valid perspective to say "we'd like to know whether this mutation is stabilising or not", unless neutral and destabilising mutations are grouped into the same category, it would require approaches for 3 categories, which are substantially more complex than the classic ROC curves.

Clustering to avoid data leakage is a good idea, however GraphPart <https://academic.oup.com/nargab/article/5/4/lqad088/7318077?login=false> recommends against MMseq2 as they still detect leakage under its usage.

A lot of the text in the supplementary is a direct copy of the main manuscript text, plus some elaboration - this is not so helpful for readers, please provide additional information only/mainly.

Reviewer #3 (Remarks to the Author):

I co-reviewed this manuscript with one of the reviewers who provided the listed reports as part of the Nature Communications initiative to facilitate training in peer review and appropriate recognition for co-reviewers.

REVIEWER COMMENTS

Reviewer #1 (Remarks to the Author):

In "STABILITY ORACLE: A STRUCTURE-BASED GRAPH-TRANSFORMER FRAMEWORK FOR IDENTIFYING STABILIZING MUTATIONS", the authors assert that their Stability Oracle pipeline surpasses other stabilising mutation predictors, citing: 1) its structure-based graph-transformer design, and 2) a generalised dataset developed from Thermodynamic Permutations (TPs) on a vast protein stability dataset. While the manuscript elaborates on the thermodynamic permutation process, ambiguities remain around the TP concept and the methodology employed.

[General]

The Thermodynamic Permutations (TP) concept as an augmentation for $\Delta\Delta G$ is both novel and intriguing. However, it raises a few questions.

The pipeline seems to focus solely on the microenvironment derived from the wild-type for structural features of all 380 TPs.

How different are those structural features across all 380 TPs? Can you give an example?

This is one of the major contributions of our method: it suffices to use a single microenvironment (often from the wildtype structure) along with individual amino acid embeddings in the transformer block of the regression head to achieve generalization. This obviates the need for a mutant structure, which is difficult (in many respects) to obtain and often not available for every mutational design during protein engineering.

To understand and demonstrate the impact of the structural features we 1) added a UMAP visualization of our T2837 to Figure 6(a) and provided a concrete example via all mutations (including TP ones) in PDB: 1JIC at position 36 (Supplementary Figure 7); 2) following the recommended standard from Kasper P. Kepp lab, we conducted a case study and compared the performance across three PDBs of P53 (Supplementary Figure 1) with visible structural differences (Supplementary Figure 1); 3) added a table demonstrating the impact on performance for the T2837 TP dataset (7720 mutations with 100% mutation type coverage) when the input microenvironment is generated via the AlphaFold structure.

Literature:

Kasper kepp paper

Notes on these three additions:

- 1) The UMAP demonstrates that the TO amino acid strongly influences the clustering, and microenvironment/structural variation appears to impact intra-cluster positioning.
- 2) This demonstrates the impact of structural changes on performance, where the protein-only dimeric structure (PDB 2OCJ) is the worst and the biologically assembled

tetramer bound to DNA structures (PDBs 3Q05 and 2AC0) results in improved performance across all metrics.

- 3) The table demonstrates that performance is mildly impacted by the local changes in the microenvironment (WT vs FROM vs TO) and generalization is preserved.

1)

Supplementary Figure 7: UMAP visualization of Stability Oracle's 128-dim subtracted hidden representation prior to decoding into a $\Delta\Delta G$. a) UMAP of T2837 colored by the "FromAA" (left) and "ToAA" (right). b) UMAP of all 380 mutation types for G36 in protein PDB: 1JIC overlaid with T2837 as background (gray). In the left and right figures, the points are colored by their "fromAA" and "toAA" amino acid type, respectively.

We have added the following text to the manuscript to address these new figures/tables in page 5, line 185-191, section 2.2:

UMAP visualization of the mutation hidden representation for T2837 from the Structural Amino Acid Embeddings reveals that to the "ToAA" CLS token drives the organization of the latent space and recovers known biochemical relationships between the 20 amino acids as illustrated in Figure 7a. We observe 1) a clustering of hydrophobes (LEU, VAL, ILE, MET), aromatics (PHE, TYR, TRP), and short polars (SER, THR and ASP, ASN) (right panel); 2) isolation of the unique amino acids (GLY, CYS, PRO) (right panel); 3) the unique situation of mutating away from GLY and adding a chiral side-chain (left panel). For a residue-specific case study of the 380 mutation types, see Supplementary Figure 7.

2)

PDB code	Resolution	biological assembly	Position	Pearson	Spearman	RMSE	AUROC	Precision	Recall	Accuracy
2OCJ	2.05 Å	homodimer	94-312	0.73	0.68	1.50	0.80	0.46	0.91	0.73
3Q05	2.40 Å	homotetramer complexed with a DNA helix	94-326	0.75	0.80	1.48	0.83	0.55	0.91	0.81
2AC0	1.80 Å	homotetramer complexed with two DNA helices	94-293	0.78	0.79	1.70	0.86	0.64	0.84	0.83
-	-	-	-	0.75±0.02	0.76±0.05	1.56±0.10	0.83±0.02	0.55±0.07	0.89±0.03	0.79±0.04

Supplementary Figure 1: Case study Stability Oracle's performance on different P53 structures (PDBs: 2OCJ, 3Q05, 2AC0). This experiment demonstrates the sensitivity of Stability Oracle to the quality of the input structure.

We have added the following text to the manuscript to address these new figures/tables in page 9, line 232-239, section 2.2:

Finally, we evaluated Stability Oracle's structural sensitivity with a case study on p53, an issue previously documented for structure-based stability predictors [69].

We evaluated three p53 structures (PDB: 2OCJ, 3Q05, 2AC0) that differ in their protein length (94-312, 94-326, 94-293), resolution (2.05, 2.40, 1.80 Å), and biological assembly (homodimer with no DNA, homotetramer complexed with a DNA helix, homotetramer complexed with two DNA helices), respectively, visualized in Supplementary Figure 1a. This case study demonstrates that Stability Oracle generalizes amid significant structural variations of p53, achieving a $Pearson=0.75\pm0.02$, $Spearman=0.76\pm0.05$, $precision=0.55\pm0.07$, and $AUROC=0.83\pm0.02$ (full results are provided in Supplementary Figure 1b).

3)

T2837 TP-Only AlphaFold Structure	Pearson	Spearman	RMSE	AUROC	Precision	Recall	Accuracy
WT	0.67	0.66	1.51	0.81	0.67	0.66	0.76
FROM	0.65	0.62	1.56	0.79	0.63	0.64	0.74
TO	0.64	0.61	1.58	0.79	0.60	0.63	0.74

Supplementary Table 4: Stability Oracle's T2837 TP-only performance on different AlphaFold structure datasets. To evaluate the impact the local structure has on generalization, we created three AlphaFold structural datasets of the TP mutations of T2837 (7720 mutations) where we varied the amino acid at the mutation position: WT, FROM, TO. Specifically, The WT, FROM, and TO datasets have the wildtype, "FromAA", and "ToAA" amino acids present at the mutation position, respectively.

We have added the following text to the manuscript to address these new figures/tables in page 8, line 221-223, section 2.2:

Next, we evaluated the impact on the "From" and "To" AlphaFold structures on the T2837 TP-only dataset (7720 mutations, 100% mutation type coverage) and observed a 2-4% drop in classification and regression metrics (Supplementary Table 4).

Would authors expect better performance if the pipeline can take the structural features from even mutants?

We did not take this approach because obtaining an accurate mutant structure is extremely difficult and not practical during the engineering of a protein. Even using AlphaFold to obtain a mutant structure would make our entire approach prohibitively expensive as it would increase the computational cost by orders of magnitude: to DMS a 300 protein we will need to obtain 5700 mutant structures (experimental or computational). That said, we **did add a supplementary table** that indicates that the choice of structure used (Alphafold structures of the WT vs FROM vs TO amino acid) only mildly impacts generalization on the T2837 TP dataset (Supplementary Table 4).

T2837 TP-Only AlphaFold Structure	Pearson	Spearman	RMSE	AUROC	Precision	Recall	Accuracy
WT	0.67	0.66	1.51	0.81	0.67	0.66	0.76
FROM	0.65	0.62	1.56	0.79	0.63	0.64	0.74
TO	0.64	0.61	1.58	0.79	0.60	0.63	0.74

Supplementary Table 4: Stability Oracle's T2837 TP-only performance on different AlphaFold structure datasets. To evaluate the impact the local structure has on generalization, we created three AlphaFold structural datasets of the TP mutations of T2837 (7720 mutations) where we varied the amino acid at the mutation position: WT, FROM, TO. Specifically, The WT, FROM, and TO datasets have the wildtype, "FromAA", and "ToAA" amino acids present at the mutation position, respectively.

To highlight the generalization across input microenvironments to the reader, we will add the following sentences in in page 8, line 221-223, section 2.2:

Next, we evaluated the impact on the "From" and "To" AF structures on the T2837 TP-only dataset (7720 mutations, 100% mutation type coverage) and observed a 2-4% drop in classification and regression metrics (Supplementary Table 4).

What features would mostly contribute to the performance of the Stability Oracle?

Stability Oracle is not a traditional machine learning algorithm that leverages a small set of human curated significant features but rather uses a graph-transformer backbone on large datasets to learn features directly from the raw atomic data of a microenvironment.

In our ablation studies, we found that the structural amino acid embeddings compared to the naive one-hot embeddings had the largest impact on performance. The innovation of structural amino acid embeddings could be considered feature engineering, which we presented in

Supplementary Table 2. We also compare different data augmentations (e.g. TR, TP) in **Supplementary Table 3.**

To make the importance of the structural amino acid embeddings more clear to the reader, we have added Figures 1c and 1d, which demonstrate how structural amino acid embeddings are input to the regression head and control the mutation directionality:

Figure 1: Overview of the Stability Oracle Framework. a) Self-supervised pre-training graph-transformer architecture (MutComputeXGT). b) Fine-tuning of the pre-trained graph transformer backbone for stability regression (Stability Oracle). In the regression head, we represent a mutation with "FromAA" and "ToAA" CLS tokens, which are the structural amino acid embeddings for the corresponding amino acid. c/d) demonstrates how Stability Oracle combines structural amino acid embeddings and one masked microenvironment to generate Thermodynamic Permutations (TP) augmentation mutation inputs. Here, $\Delta\Delta G$ measurements at PDB:5UCE W43 (yellow transparent spheres) for mutations to both LEU and ARG enable the generation of the TP mutations c) from LEU to ARG and d) from ARG to LEU by simply swapping the order of the structural amino acid embeddings provided to the regression head. Diagram further describing TP is provided in Supplementary Figure 2.

Regarding the concept of TPs, for example, if Y->A results in $\Delta\Delta G_1$ and Y->D yields $\Delta\Delta G_2$, the computed difference, $\Delta\Delta G_2 - \Delta\Delta G_1$, might imply a D->A mutation. Yet, if one were to adhere to the thermodynamic cycle, the sequence would be D->Y->A. This indicates that the true effective mutation can be Y->A rather than D->A. In this sense, is it correct that the application of TP seems restricted to the transformer attention layer, particularly the "from" and "to" segments, given that the structural disparity between wild and mutant forms remains unchanged for both TP and TR?

Yes, it is correct that the application of TP is restricted to the regression head. Subsequent to the transformer attention layer we subtract the structural amino acid embedding CLS tokens. This architectural design was intentional to 1) minimize computational cost (one structure); 2) enforce mutational symmetry via subtraction of AA CLS tokens ($AA_1 - AA_2 = \Delta\Delta G_1$ & $AA_2 - AA_1 = -\Delta\Delta G_1$); 3) enable mutational permutations (using FromAA and ToAA CLS tokens as input to the regression head and not the graph transformer backbone).

The input structure does impact performance but the impact is minor and does not compromise generalization. We demonstrate this by comparing the performance on all T2837 TP mutations (7720 mutations) using the AlphaFolded structure of the WT, FROM, or TO amino acids (in contrast to experimentally derived structures). This experiment is provided in Supplementary Table 4 and demonstrates that the effect is minor and does not compromise generalization. These results have been discussed and presented in several of the previous questions.

[Others][P2, 39] Rosetta is a physical-based not a knowledge-based method.

We thank the reviewer for the correction. The text has been corrected in page 2, line 38, section 1:

“The community still primarily relies on physics-based methods,”

[P2, 42-43] What is the criteria of the data leakage?

The criteria we are using for data leakage is the one defined by Rost, 1999: proteins below 30% sequence similarity are considered to be in the “twilight zone” and there is 95% chance they are not within the same protein fold. This threshold is also used by the GraphPart algorithm suggested by reviewer 2.

We have modified the text later in the introduction to define data leakage in page 2, line 74 - 77, section 1:

“We present three new datasets (C2878, cDNA117K, T2837) to address data leakage issues. MMseqs2 [46] was used to generate these three datasets and ensure the maximum overlap between proteins in the training and test set is below 30% sequence similarity, a threshold within the ‘twilight zone’ where 95% of protein pairs will have different structural folds [47].”

[P2, 47-49] The author highlighted the challenges of using regression metrics on unbalanced datasets, a characteristic intrinsic to this type of dataset. However, while the MCC addresses imbalance, it doesn't capture the predictive accuracy of regression tasks, specifically the precision in predicting $\Delta\Delta G$ values. Also, based on the results, Pearson's correlation coefficients showed reasonably similar to Spearman's rho, even for non-TP test datasets.

We agree with the reviewer that MCC does not capture the predictive accuracy but does address imbalance. Thus, we remove the word “prioritized” and instead use “should also be considered” to emphasize the importance of both imbalance classification and regression metrics for ML-guided protein engineering applications.

The text in page 2, line 51 - 53, section 1, now reads:

Thus, metrics such as precision, recall, area under receiver operating characteristic (AUROC), and Matthew's correlation coefficient (MCC) should also be considered when developing a model for protein engineering applications [13,14,31].

[P2, 53-58] Does the Stability Oracle pipeline produce all 380 potential TP cases from just one wild-type structure? If so, does the platform solely incorporate structural features

from the wild-type PDB structures? While this approach might address the scarcity or bias of $\Delta\Delta G$ labels, it may not fully account for variations in structural features.

Yes, the Stability Oracle pipeline was designed to produce all 380 mutation types from just a single structure, which is often the wildtype. Thus, it indeed solely incorporates structural features from a single structure and does not **explicitly** account for structural variations induced by a mutation, but rather is forced to learn to extrapolate such mutational structural changes. We accomplish this by masking the mutated residue in the microenvironment and pre-training the model to predict the masked WT amino acid across 2.3M microenvironments. We hypothesize that this pre-training enables the feature extractor **to learn meaningful structural variations** and enables the classifier to learn how each amino acid favors and disfavors these local variations (the source of structural amino acid embeddings). Then, during stability fine-tuning, we build on the pre-trained masked microenvironment feature extractor and convert the structural amino acid embeddings into CLS tokens to represent the FROM and the TO amino acids in a mutation. The rationale for this was to allow the model to **infer** the changes in the local structure rather than **explicitly model** them with a second computationally generated structure.

To make this technique more clear to the reader, we added two subfigures to Figure 1 (1c and 1d), which demonstrate the “From” and “To” input for TP generated mutations (structure AA: TRP, c) From: LEU, To: ARG; From: ARG, To: LEU), respectively. Figure 1 (1c and 1d) have been demonstrated in previous questions.

[P2, 65-66] My comprehension is that TP is introduced when a residue has multiple mutational $\Delta\Delta G$ values. For instance, if there are two mutational $\Delta\Delta G$ values, such as $\Delta\Delta G$ Y->A and $\Delta\Delta G$ Y->K, then we can derive $\Delta\Delta G$ K->A and $\Delta\Delta G$ A->K using TP. However, Alanine Scanning results in a single mutation for each residue (X->A), making it unsuitable for the TP process.

Yes, if a residue only has experimental results from Alanine Scanning then TP does not attenuate this. The augmentation power of TP comes from when there are multiple amino acids experimentally characterized at the same residue. However, TP does attenuate the Alanine bias across the datasets by lowering its relative prevalence. This is demonstrated with the heat maps in Figure 3, middle and right column, where C2878 and T2837 still have an alanine bias but it has been dampened.

[P2, 69-71] Did the author suggest that data leakage can be effectively managed by limiting sequence similarity between the training and test sets to 30%? In the introduction, the authors highlighted that data issues are prevalent and difficult to address using methods like clustering. Therefore, a comprehensive explanation is warranted. Is there empirical evidence to confirm that the 30% threshold is ideal for preventing data leakage?

We thank the reviewer for bringing this ambiguity to our attention. The reason why we chose to use a 30% sequence similarity is that it is well established that proteins with sequence similarity below this threshold enter the “twilight zone”, a phenomenon where ~95% of protein pairs have different structure folds (Rost 1999). Thus, providing us confidence that our proteins are structurally dissimilar. Furthermore, this threshold was also used by the GraphPart algorithm recommended by Reviewer 2.

To clarify this to readers, we have modified the text page 2, line 74 - 77, section 1:

“We present three new datasets (C2878, cDNA117K, T2837) to address data leakage issues. MMseqs2 [46] was used to generate these three datasets and ensure the maximum overlap between proteins in the training and test set is below 30% sequence similarity, a threshold within the “twilight zone” where 95% of protein pairs will have different structural folds [47].”

Literature:

Rost, Burkhard. "Twilight zone of protein sequence alignments." *Protein engineering* 12.2 (1999): 85-94.].

[P2, 73-74] How many mutations are there in natural versus de novo proteins?

In our preprocessing of cDNA, we only use natural proteins and exclude all data from de novo proteins. We made this decision to ensure we are training with the stability data of **functional** proteins. We know very little about the de novo domains and their function outside of folding correctly. This will be clear in the open-sourced dataset we will deposit in the github repo.

We have clarified this in page 2, line 78 - 80, section 1:

“The third (cDNA117K) is a curated subset of **the natural domains from** the cDNA display proteolysis dataset #1 [48], a dataset of ~850K thermodynamic folding stability measurements across 354 natural and 188 de novo mini-protein domains (40-72 amino acids).”

We have clarified this in the cDNA methods in line 661, supplementary information B.2:

“First, **we removed all de novo domains** and then filtered all single point mutations on protein scaffolds that have a wildtype structure pdb id provided in the csv.”

[P2, 77] I'm unable to determine what "Prostata-IFML" is and how it differs from "Prostata." Could you provide more information or clarification?

Prostata is an ensemble of 5 models with 5 different regression head architectures for processing the ESM2 residue embeddings of a mutation. Prostata-IFML is a single model that uses the outer-product regression head architecture proposed in the Prostata paper that we fine-tune on our training/test splits. Essentially, Prostata-IFML is our version of fine-tuning

ESM2's embeddings on our dataset splits; we credit the original authors for developing the outer-product regression head. We mention the details in supplementary B.3.

We have clarified this in page 10, line 274-281, section 2.4:

“Prostata [26] is a sequence-based framework that fine-tunes ESM2 embeddings **ensembling five distinct regression heads** (a SOTA protein language model) on common training and test sets. **However**, Prostata was trained with **homologous proteins** (a sequence similarity cutoff of 75%) with respect to SSym and S669, **resulting in** inflated performance on T2837 and its subset test sets (breakdown of the performance and data leakage are provided in Supplementary Table 4). In order to address this data leakage and conduct a fair comparison, we fine-tuned ESM2's representation using the same training and test sets as Stability Oracle **and only the outer-product regression head architecture**.

[P2, 86-88] Do the authors suggest that self-supervised models produce unbiased models even on uneven datasets?

We explain that the self-supervised models can serve as zero-shot learning models for different tasks, in which the label information (here it is $\Delta\Delta G$) is never seen in the self-supervised training.

To specifically answer your question, we do not suggest that self-supervised models produce unbiased zero-shot models but rather by leveraging self-supervised pre-training on millions of microenvironments, we mitigate the biases present in scarce experimental datasets (such as alanine biases and missing mutation types), improving the generalization of a fine-tuned model.

[P3, 109] AlphaFold is better suited for modeling general proteins rather than specific point mutants. (“Can AlphaFold2 predict the impact of missense mutations on structure?”)

It has been demonstrated that AlphaFold is not guaranteed to generalize to structural variation induced by single-point mutations accurately (Pak et al. 2023). We are not aware of a computational tool that can successfully model structural changes from single point mutations. This is another reason why we designed Stability Oracle to use a single structure as input and not rely on a second structure that is often computationally generated with questionable accuracy.

Literature:

Alphafold is not sensitive to mutations (Pak, Marina A., et al. "Using AlphaFold to predict the impact of single mutations on protein stability and function." *Plos one* 18.3 (2023): e0282689.)

[P4, Figure 2] No explanation for FP.

Thank you for pointing this out. We added the following explanation in Figure 2 caption:
“Q1744, O2567, and FP (FireProtDB [63]) represent different datasets.”

[P5, 161-163] Closing parenthesis is missing

Thank you for pointing out this syntax issue. We added missing parentheses.

[P5, 174] Figure 4 should come earlier than 5

We agree with the reviewer and have changed the order.

[P5, 180] Is the C2878 + TR correct? or should it be C2878 + TP?

We thank the reviewer for pointing out our poor wording. Yes, it is correct. Here, our point is that using the traditional augmentation technique (TR) is less effective at addressing missing mutation types compared to TP, our technique. We demonstrate that with respect to the 12 mutation types unaddressed by C2878 + TR, TP augmentation of the dataset can improve generalization.

To clarify this point, we have modified the text in page 8, line 196-198, section 2.2:

To further examine how TP-augmented datasets impact generalization, we evaluated predictions at the 12 mutation types that only have experimental data available when using TP (C2878 + TP) and not when using the traditional augmentation technique TR (C2878 + TR) (see Supplementary Figure 3, row 1, column 3 off-diagonal blue squares for the missing mutation types).

[P5, 181-182] In the test data (T2837 + TP), there are 12 mutation types (e.g., Ala to Tyr) that are not present in the C2878 training dataset, nor in the combined C2878 + TP dataset. Is that correct? Can authors describe what are the 12 mutation types?

We thank the reviewer for highlighting this confusion, the 12 mutation types are not present in C2878 + TR. However, Thermodynamic Permutations of C2878 (C2878 + TP) is a data augmentation that makes these 12 mutation types available during training. We have updated Figure 5 caption to reference the Supplementary Figure 3 and explicitly list them in the figure caption.

The 12 mutation types have been added to the supplementary figure 3 for the reader:

“Supplementary Figure 2: Heatmap of C2878, cDNA117K, and T2837. The first column shows the mutation type distribution of the original data, the middle column shows the TR augmented data, and the right column shows the original + TR datasets. For C2878 + TR, the 12 missing mutation types are: R->C, C->H, C->K, C->M, C->P, Q->W, C->R, H->C, K->C, M->C, P->C, W->Q. For T2837 + TR, the 10 missing mutation types are: W->N, P->C, W->P, W->T, T->W, P->W, C->P, N->W, K->L, L->K.”

[P5, 182-183] The authors described “...observed similar results”, but there is a notable difference in performance between using (0.28 -> 0.40) and not using (0.58 -> 0.62) a filter for the training dataset. Additionally, why wasn't the MCC metric included, especially considering the authors' primary assertion that the three metrics ("Accuracy," "Recall," and "Precision") are not reliable?

To clarify our assertions: we assert that traditional regression metrics, such as Pearson correlation, error (RMSE), and accuracy, are not reliable to guide model development, especially when trying to develop a framework for predicting stabilizing mutations. Here, we primarily focused on precision and recall since these metrics are specific to stable mutations (the positive class), which is the primary goal for most protein engineering applications, while MCC balances TP and TN performance. As for the recall comparisons, we agree that in isolation 0.58->0.62 is much smaller than 0.28->0.40. However, improvement in recall is usually associated with a trade-off in precision. Here, however, TP enables the 0.58->0.62 recall improvement with a simultaneous precision improvement of 0.49->0.56, highlighting the overall effectiveness of TP for improving generalization of mutation types with no experimental data available.

We modified the text to highlight the attenuated effect with artificially increased mutation types in page 8, line 200-201, section 2.2:

“We artificially expanded the mutation types that were missing data to 54 and observed similar but attenuated improvements to both precision and recall (Figure 5).”

[P5, 188] It's somewhat unclear; I'd suggest sticking with a 30% sequence homology for clarity.

We thank the reviewer for pointing out how unclear our phrasing was.

We have clarified the text in page 8, line 207-209, section 2.2 from:

“Our analysis reveals a 100% homology overlap (>30% sequence similarity threshold) between the proteins in cDNA120K and C2878, providing a rationale for the lack of performance improvement.”

To:

“Our analysis reveals all proteins in C2878 are homologous (>30% sequence similarity) to at least one protein in cDNA117K and therefore C2878 does not expand the protein space available for training. This observation provides a rationale for the lack of performance improvement observed upon further fine-tuning on C2878.”

[P7, 198-200] Could the observed behavior be because the Stability Oracle does not rely on structural features as much? What do the authors think about this?

We believe the reason for the observed strong generalization using AlphaFolded structures is due to the fact that only well-characterized proteins possess $\Delta\Delta G$ measurements. Thus, all proteins in T2837, and their corresponding scaffold/fold, most likely had a homolog in the AlphaFold training set. Even though we used template-free AlphaFold to generate our structures, the structure quality was very good, and similar results between experimental and AlphaFold structures were expected.

[P8, Figure 6] Could the authors consider modifying the symbol representations for clearer visual presentation?

After further examination of the figure, we have concluded that the ThermoNet and Rosetta symbols are too similar. We have changed the symbol and color representation of ThermoNet and Rosetta to be more distinct from each other, made the PROSTATATA-IFML symbol in P53 bigger than the previous one, and changed the color for RaSP to purple.

Furthermore, to help the reader compare all the literature models, we added a table version of the figure as Supplementary Table 11 to facilitate comparison.

Figure 6: The Pearson correlation coefficient of Stability Oracle and Prostata-IFML across several test sets. We compare against a handful of computational stability predictors from the community (values obtained from the literature and are also provided in Supplementary Table 11 [34, 36, 38, 70, 71]).

Method	P53 (2OCJ)		Myoglobin (1BZ6)		S-sym		S669		T2837	
	Forward	Inverse	Forward	Inverse	Forward	Inverse	Forward	Inverse	Forward	Inverse
Stability Oracle	0.73	0.73	0.68	0.68	0.72	0.72	0.52	0.52	0.59	0.59
PROSTATATA-IFML	0.73	0.73	0.55	0.55	0.56	0.56	0.51	0.51	0.53	0.53
ThermoMPNN	0.66	0.57	0.58	0.49	0.66	0.58	0.40	0.34	-	-
ThermoNet	0.45	0.56	0.38	0.37	0.47	0.47	0.39	0.38	-	-
ProsGNN	-	-	0.48	0.43	0.61	0.56	-	-	-	-
DDGUN	-	-	-	-	0.48	0.48	0.41	0.38	-	-
PremPS	-	-	-	-	-	-	0.41	0.42	-	-
Rosetta	-	-	-	-	0.68	0.43	0.39	0.40	-	-
DynaMut	-	-	-	-	-	-	0.41	0.34	-	-
SDM	-	-	-	-	-	-	0.41	0.13	-	-
FoldX	-	-	-	-	0.63	0.39	-	-	-	-
RaSP	0.67	0.10	0.68	0.36	0.64	0.30	0.39	0.27	0.55	0.23
DUET	-	-	-	-	0.63	0.13	0.41	0.23	-	-

Supplementary Table 11: Pearson correlation coefficient (PCC) comparison of Stability Oracle and Prostate-IFML against literature computational stability predictors.

[P8, 220] The metrics used for the Y-axis in Figure 10 and those in Table 5a are unclear.

We Thank the reviewer for highlighting this to us. We have added y-axis labels to Figure 5 and Figure 10.
Figure 5

Figure 10:

[P8, 220-222] Is the Stability Oracle's success rate for stabilising mutations 48.1%? If that's the case, it doesn't outperform the "FEP methods" which are around 50%. Moreover, if "success rate" is synonymous with accuracy, wouldn't using the MCC provide a more precise measure?

We thank the reviewer for pointing out the confusion here. The success rate of Stability Oracle is 74% (precision score): from all the stabilizing mutations predicted by Stability Oracle, 74% indeed are experimentally stabilizing mutations. While 48.1% is the recall score: Stability Oracle predicts 48.1% of all experimentally stabilizing mutations in the dataset. FEP methods are computationally expensive, thus, it is impractical to calculate their recall due to the need of having to run FEP on every stabilizing mutation in the dataset. The FEP success rate (~50%) is the percentage of mutations calculated to be stabilizing that are actually experimentally stabilizing, which is akin to precision. Thus, while we do not conduct a comprehensive comparison between the two methods, we conclude that the precision of Stability Oracle *may surpass* the precision of FEP while being orders of magnitude less computationally intensive.

We believe MCC is the wrong metric here because it takes into account both false positives and false negatives. Since we cannot obtain a recall (false negative) metric for FEP (described above), we are unable to compare MCC values between Stability Oracle and FEP methods.

We will clarify this by adding the following text in page 9, line 254-256, section 2.3:

“The success rate of predicting stabilizing mutations (74%) appears to surpass what is typically observed with FEP methods (~50%) [31, 66] with orders of magnitude less computational cost (Supplementary Table 1).”

[P8, 220-222] Also, it would be more informative if the Y-axis represented the TP rate instead of Counts, facilitating a smoother comparison of performance across different amino acids (AAs).

The reason why we chose to use counts rather than true positive rate is due to the underlying mutation type bias in the T2837 + TP dataset. While TP does alleviate the bias, it is still a significant issue when it comes to evaluating performance on the 380 mutation types and concluded that normalization would mislead the reader with inflated Stability Oracle's performance on mutation types with limited data. This bias is explicitly shown in supplementary figure 4 for the 380 mutation types.

[P9, Figure 8] The authors referred to the performance difference as insignificant, but how does this apply to the case of T2837, especially for Recall RSA within the range of 0.1 to 0.3?

We would like to thank the reviewer for highlighting this confusion. We do the test of significance on the entire dataset and not individual RSA bins because there is not enough data to do a test of significance per RSA bin.

We will clarify this by adding the following text in Figure 8 caption:

“The results demonstrate that there are no biases between polar and hydrophobic residues throughout a protein structure on both T2837 and T2837 + TP: we do a test of significance for the absolute value **between** polar and hydrophobic data examples on **the whole dataset (not per RSA bin due to lack of data) and it is insignificant.**”

[P27, 651] Same as earlier, Rosetta is a physics-based approach.

We thank the reviewer for bringing this to our attention. We have corrected this in the manuscript in page 33, line 722,, supplementary information C:

“The community still primarily relies on **physics-based** methods, such as Rosetta.”

[P27, 660-662] Same as earlier, AlphaFold2 may not be optimal for generating single-point mutants. Approaches based on homology modelling might be more effective. If the authors recommend using AlphaFold2/Rosetta over MODELLER or PyMOL for mutagenesis, there should be justification highlighting the superior performance of AlphaFold2/Rosetta.

We thank the reviewer for pointing out our accidental bias towards AlphaFold/Rosetta. We do not recommend using AlphaFold2, Rosetta, Modeller, PyMol, or any other tool for mutagenesis since they are all computational tools trying to model a highly dynamic microenvironment around a single-point mutation. We simply mention these two since they are the most prevalent methods used by other frameworks that require a “From” and “To” structure as input, e.g. ThermoNet, Pros-GNN, FoldX, etc.

To reflect this, we have made the following changes in page 33, line 731-733, supplementary information C:

“For example, to conduct a computational deep mutational scan (DMS) of a 300 amino acid protein, one must computationally generate 5700 mutant structures, **with a AlphaFold2 [22] or Rosetta [32] being the most prevalent tools used in the literature.**”

[P28, 697] Could you clarify what "game-ability" refers to in this context?

Here, game-ability refers to the ease of modifying the model's features, hyperparameters, and/or datasets to improve its performance on poorly constructed validation/test sets. This leads to researchers publishing overfit results on their validation set at the expense of poor generalization when examined by third-party researchers.

We have modified the text to better contextualize our explanation in page 34, line 767 - 771, supplementary information C:

“This misalignment has resulted in metrics that do not prioritize improvement in accurately predicting stabilizing mutations and their game-ability **constantly results in inflated reported performance metrics that fail to generalize upon evaluation by**

third-party researchers. This game-ability and in turn poor generalization for stabilizing mutations is demonstrated in great detail in a study that consisted of 21 established computational tools [31].”

[P28, 699] Could you clarify what “overly simplified model” is in this context?

We added a line to describing the overly simplified model paraphrasing the original publication text, page 34, line 773 - 776, supplementary information C:

“The overly simplified model used was an artificial neural network with a single, two-neuron hidden layer that takes four commonly used structural descriptors of amino acid substitution as input: change in volume, change in hydrophobicity, change in secondary structure propensity, and location via solvent exposure.”

[P28, 698-700] The authors utilised the "Spearman" and "MCC" metrics, but they have not provided a clear comparison to demonstrate their superiority over the original metrics.

This section is intended to be a high-level overview of the current issues associated with the metrics used by the computational stability prediction community. Thus, we are summarizing the conclusions of an in-depth analysis of over 20 computational stability prediction tools. A clear thorough comparison would be out of the scope of the section. However, in this manuscript, we prioritized precision, recall, and AUROC to guide our model development.

To make this more clear, we have made the following changes to the manuscript, page 34, line 775 - 778, supplementary information C:

“For a more in-depth metric comparison, we refer the reader to [31]. We argue that these metrics are still insufficient for monitoring stabilizing mutation predictive improvement. Here, we use classification metrics, such as precision, recall, and AUROC to monitor improvements for identifying stabilizing mutations during the development of Stability Oracle.”

Reviewer #2 (Remarks to the Author):

In “STABILITY ORACLE: A STRUCTURE-BASED GRAPH-TURNFORMER FRAMEWORK FOR IDENTIFYING STABILIZING MUTATIONS”, Diaz et al. Present a new method for the prediction of stability changes in proteins upon mutation. This is a longstanding field with a plethora of known methods, and while the manuscript presents some interesting new ideas, there are also a number of issues that need to be addressed.

There’s a number of issues in relation to the existing literature that makes one wonder if the authors misunderstand the general goal of stability predictions in this field. The most common effect of a mutation on a protein is loss of stability (not gain), and this is a very valuable insight in many areas from biotechnology to interpretation of pathogenic

variants. Thus, the perceived discrepancy in lines 40-41 is largely due to the fact that most variants are destabilising, and that these are often predicted correctly. This also explains the imbalance in test/train sets. These are all well-known and long-standing issues - they are properties of (naturally evolved) proteins.

We thank the reviewer for highlighting this key distinction between the established stability prediction community and the emerging machine learning-guided protein engineering community.

We have rewritten the text to clarify the distinction between the two communities in page 2, line 40 - 47, section 1:

Original:

“Furthermore, third-party analyses emphasize that only about 20% of predicted stabilizing mutations from the current state-of-the-art computational tools are actually stabilizing, even though 75-80% accuracy is often reported upon publication [14, 31, 13]. This poor generalization to new datasets highlights how pervasive the data leakage between train-test splits is in the computational stability prediction community [16, 31, 14, 13].”

Updated:

While current state-of-the-art computational tools often report 75-80% accuracies upon publication, these accuracies primarily reflect their performance on destabilizing mutations, which make up a majority of the test set and their identification is key for identifying pathogenic variants [13, 14, 31]. However, when evaluated for predicting stabilizing mutations on new datasets by third parties, about 20% of predictions are actually stabilizing [13, 14, 31]. Although stabilizing mutations are naturally less common [43], poor generalization to stabilizing mutations has been demonstrated to be a result of pervasive data leakage between train-test splits and severe class imbalance (destabilizing mutations make up >70%) in the current training sets used by the computational stability prediction community. [13, 14, 16, 31].

Along similar lines, the authors state on p3 that "previous structure-based prediction programs need 2 input structures, mutant and wild-type" - this is either wrong or at least misleading. The cited papers are from 2019-2020, while structure-based stability prediction has been around since the early 2000s at least (Guerois et al. 2002; Kortemme et al. 2004; Kellogg et al. 2011). All these tools use one input structure, the other is generated - and the structural accuracy of these mutant structures (or lack thereof) is a longstanding issue and not taken lightly.

We would like to thank the reviewer for pointing us to additional structure-based computational tools, which we will cite, and highlighting the confusion surrounding the word “structure”. We agree with the author that all the tools from 2002-2011 and from 2019-2020 use one input

structure (usually the experimental structure of wildtype) and a second generated structure for the mutant. However, an experimental wildtype structure and a computationally generated mutant structure are still two structures. In our manuscript, when we refer to a structure (whether it's one or two) we are agnostic to the method of generation (Experimental: X-ray, CryoEM, NMR, etc. Computational: Rosetta, AlphaFold, Modeller, PyMol, Chimera, etc) and simply mean a .pdb or .cif file with atomic coordinates of a protein. Thus, it is unclear to the authors how "previous structure-based prediction programs need 2 input structures (wildtype and mutant) is "wrong or misleading" since we both agree that all tools from 2002-2011 and from 2019-2020 use two structures as the input—Quoting the reviewer: "All these tools use one input structure, the other[structure] is generated".

Unlike the previous tools, our method uses just one input structure (whether computational or experimental) as the input and does not require a second structure to model the mutant (whether computational or experimental). Instead we augment this one structure with representations/encodings of the from and to amino acids. We call these encodings "Structural Amino Acid Embeddings", and they significantly improve generalization.

To clarify this in the manuscript, we have added Figure 1c and 1d and made the following changes to the text in page 3, line 110-115, section 2.1:

Original:

"Previous structure-based stability predictors [28, 29] require two structures as inputs, one for the wildtype and one for the mutant. Instead, Stability Oracle uses only one structure encoded via a MutComputeXGT representation, and two amino acid level embeddings, one for the "from" and one for the "to" amino acid."

Updated:

"Previous structure-based stability predictors [28, 29, 56, 57, 58] require two structures (experimental and/or computational) as inputs to *explicitly* model the wildtype and mutant amino acids, respectively. Instead, Stability Oracle uses only one structure (experimental or computational) to model the local chemistry (masked microenvironment), which is embedded via the MutComputeXGT feature extractor and models the mutation "from" one amino acid "to" another amino acid *implicitly* via embedding vectors (as illustrated in Figure 1c and 1d)."

This leads into the second major issue: in suppl data A.3 - data augmentation the authors illustrate how they combine mutations in some way to generate additional input/training data, called "thermodynamic permutation". While in principle this is possible, it should be described in specific examples how this is carried out.

While we provide an abstract diagram that delineates both thermodynamic reversibility and thermodynamic permutations (supplementary figure 1), we agree with the reviewer that a specific example will simplify the understanding of thermodynamic permutations. Thus, we have added this in the new figures, e.g., Figure 1c and 1d, demonstrating a forward and reverse

mutation generated by thermodynamic permutations. Here, the masked wildtype amino acid for the microenvironment is TRP, which is shown as semi-transparent orange spheres. Experimental characterization of mutations to LEU and ARG enables prediction of mutating “from” LEU “to” ARG and the reverse mutation (“ARG” -> “LEU”) simply by switching the order of the structural amino acid embeddings for LEU and ARG.

Figure 1: Overview of the Stability Oracle Framework. a) Self-supervised pre-training graph-transformer architecture (MutComputeXGT). b) Fine-tuning of the pre-trained graph transformer backbone for stability regression (Stability Oracle). In the regression head, we represent a mutation with “FromAA” and “ToAA” CLS tokens, which are the structural amino acid embeddings for the corresponding amino acids. c/d) Demonstrates how Stability Oracle combines structural amino acid embeddings and one masked microenvironment to generate Thermodynamic Permutations (TP) augmentation mutation inputs. Here, $\Delta\Delta G$ measurements at PDB:5UCE W43 (yellow transparent spheres) for mutations to both LEU and ARG enable the generation of the TP mutations c) from LEU to ARG and d) from ARG to LEU by simply swapping the order of the structural amino acid embeddings provided to the regression head. Diagram further describing TP is provided in Supplementary Figure 2.

The main questions though are - how do the authors have access to starting structures for these permutations? A challenge in symmetrising classic ddG approaches is that the very starting structures for the Mut -> wt direction are missing.

We agree with the reviewer that access to the starting structures to conduct thermodynamic permutations or classic symmetry ddG approach (Mut -> WT) is very challenging since mutant structures are almost always missing. One of our major contributions is to overcome this problem! As described above, we do this by using a single structure (whether experimental or computational) that masks the residue being mutated and **implicitly** models the two amino acids for the residue being mutated with **embeddings** (fromAA -> toAA: which could be wt -> mut, mut -> wt during TR, or mut1 -> mut2 during TP), rather than **explicitly** modeling the mutation with two structures each with a different amino acid. Thus, we have a “FromAA” embedding and a “ToAA” embedding to represent the specific mutation type. The naive embedding is the one-hot encoding of the 20 amino acids. Here, we propose “structural amino acid embeddings” which significantly outperform the naive one-hot embeddings (shown in Supplementary Table 2). Thus, by 1) masking the mutated residue in the structure and simultaneously 2) representing a mutation with embeddings, we circumvent the need to access/generate the mutant structure to conduct thermodynamic permutations or the classic symmetry ddG. This is one of the primary contributions of this paper. We visually depict this in the added Figure 1c and 1d, previously described.

It would not be fair to use structures generated by the tool itself, reverting mutations are far easier to predict. There are sets with both wt and mutant structures, but they are far smaller, definitely too small for deep learning. Even AlphaMissense does not predict structures (just scores), which may serve as another indication that this is a hard problem. Further, even if structures are available, the local environment of these mutations is going to be extremely similar, hence the actual increase in information will not be massive. It would mainly serve to achieve symmetry.

We would like to clarify to the reviewer that Stability Oracle does not generate structures but, much like AlphaMissense, just ddG scores. Additionally, we agree with the reviewer that the set of wildtype and mutant structures that have been experimentally solved are far too small for the training of deep neural networks and building frameworks that depend on mutant structures is not conducive for the advancement of the machine learning-guided protein engineering community.

The currently best-performing tool is ThermoMPNN <https://doi.org/10.1101/2023.07.27.550881> - the authors need to report performance compared to that.

During the writing of this paper, ThermoMPNN was not on the preprint archive. Subsequently, in the ThermoMPNN paper the authors compare against Stability Oracle. Aside from the forward Pearson on SSym (which ties our result of 0.72), we outperform them on all established test sets for both Pearson and RMSE for forward and reverse (especially in the reverse case, which their model is unable to generalize to). This is explicitly written in their paper.

Furthermore, we conducted comparisons ourselves by running their publicly available model (see <https://github.com/Kuhlman-Lab/ThermoMPNN/tree/main/models>) on the literature test sets (P53, Myoglobin, Ssym, and S669) reporting Pearson, spearman, RMSE, and AUROC in supplementary table 10. Our results demonstrate that ThermoMPNN performance on these literature test sets does not match what they report in their preprint. Thus, we report both the numbers in their preprint and those we obtained locally. Stability Oracle outperforms ThermoMPNN on nearly every metric on every dataset in the forward direction and for every metric on every dataset in the reverse direction.

Metric	Method	P53 (2OCJ)		Myoglobin (1BZ6)		S-sym		S669	
		Forward	Inverse	Forward	Inverse	Forward	Inverse	Forward	Inverse
Pearson	ThermoMPNN ¹	0.76	-	0.60	-	0.72	0.60	0.43	-
	ThermoMPNN*	0.66	0.57	0.58	0.49	0.66	0.58	0.40	0.34
	Stability Oracle	0.73	0.73	0.68	0.68	0.72	0.72	0.52	0.52
Spearman	ThermoMPNN ¹	0.70	-	0.60	-	-	-	-	-
	ThermoMPNN*	0.61	0.54	0.58	0.50	0.64	0.55	0.41	0.34
	Stability Oracle	0.68	0.66	0.66	0.66	0.70	0.70	0.53	0.53
AUROC	ThermoMPNN*	0.76	0.73	0.77	0.74	0.83	0.78	0.72	0.69
	Stability Oracle	0.80	0.80	0.82	0.81	0.87	0.87	0.75	0.75
RMSE	ThermoMPNN ¹	-	-	-	-	1.12	1.53	1.52	-
	ThermoMPNN*	1.49	1.78	0.95	1.04	1.17	1.59	1.56	1.84
	Stability Oracle	1.50	1.51	0.90	0.89	1.22	1.19	1.43	1.43

Supplementary Table 10: Comparison with ThermoMPNN. ThermoMPNN¹ is the performance reported in the ThermoMPNN preprint and ThermoMPNN* is performance we obtained from the model checkpoint in their official

GitHub repository: <https://github.com/Kuhlman-Lab/ThermoMPNN/tree/main/models>. We did not evaluate the extent of data leakage with these test sets for ThermoMPNN

We have updated the text to report this experiment in page 11, line 299 - 303, section 2.4:

In parallel to this work, ThermoMPNN--a deep learning framework that fine-tunes the ProteinMPNN [85] representations also on the megascale cDNA proteolysis dataset--was developed. Using the publicly-available checkpoint, we found that Stability Oracle outperforms ThermoMPNN on SSym, S669, myoglobin, and P53 across multiple regression and classification metrics (Supplementary Table 10).

RaSP predicts ddGs in the (-1,7) range and is therefore not really made for stabilising mutations, the focus is on destabilising ones. Hence its lower performance in the benchmarks is not surprising.

We agree with the reviewer that RaSP's lower performance on stabilizing mutations is not surprising since it was designed for the established stability prediction community and Stability Oracle was designed for the machine learning-guided protein engineering community.

However, we would like to highlight that on a primarily destabilizing test set (T2837) RaSP achieves an AUROC of 0.61 compared to Stability Oracle achieving 0.81. Highlighting the overall improved classification performance on a primarily destabilizing test set, which was not the intent behind developing Stability Oracle.

Further, the most accurate ddG dataset is that on proteinG (Nisthal et al. 2019), which the authors should also benchmark on

We thank the reviewer for pointing us to ProteinG. We did an analysis on the proteinG dataset and made sure we had no overlap in our training set with it (30% sequence similarity). Results have been added as a supplementary table 6. Stability Oracle achieves a Pearson of 0.75 with all 935 mutations and a Pearson correlation of 0.67 on the quantitative subset (830 mutations).

Dataset	# Mutations	Metric	Stability Oracle	Stability Oracle*	Prostata-IFML	Prostata-IFML*	RaSP
All	935	Pearson	0.75	0.71	0.78	0.66	0.74
		Spearman	0.71	0.66	0.79	0.64	0.64
		AUROC	0.84	0.82	0.90	0.82	0.66
		RMSE	1.68	1.69	0.95	1.16	1.42
Quantitative	830	Pearson	0.68	0.64	0.75	0.58	0.60
		Spearman	0.62	0.59	0.76	0.58	0.53
		AUROC	0.81	0.80	0.89	0.80	0.65
		RMSE	1.51	1.54	0.76	0.82	1.33

Supplementary Table 6: Regression and Classification performance of Stability Oracle, Prostata-IFML and RaSP on G β 1 (PDB:1PGA). * indicates that we retrained on the cDNA117K dataset with PDB:1YU5,1GJS,5UBS,5UCE removed in order to address data leakage (>30% sequence similarity) with G β 1. We report results on all 935 mutations (qualitative + quantitative) and the 835 mutations quantitative subset as reported in [43]. RaSP was not retrained to address any potential data leakage.

We have added these additional results in the text in page 9, line 229 - 232, section 2.2:

To date, the most accurate and exhaustive thermodynamic stability dataset in the literature is the G β 1 dataset [43]. We evaluate Stability Oracle's performance on G β 1 [43] and, to the best of our knowledge, achieve SOTA on all 935 mutations (Pearson=0.75, AUROC=0.84) and the 835 mutation quantitative subset (Pearson=0.67, AUROC=0.81) (full results are provided Supplementary Table 6).

line 69: for "new test sets", it should be very clear what were the original data that went into this, how many sequences there were, how many were removed. Also the new datasets should be provided - the github repository is empty!

We fully intend to make the datasets public and will update the GitHub repo with the datasets that went into training Stability Oracle upon publication.

A.6 predicted stabilising mutations - many of these are from Gly to hydrophobic residues, based on experience this can be a massive source of error, have the authors tried to validate this in any way?

We agree with the reviewer that experimental validation of ML-guided designs requires human review/filtering. The results shown in Supplementary Figure 7 are based on the predictive performance on T2837 + TP. We did not examine specific mutation types when preparing the manuscript but we assume the experimental data that make up T2837 are accurate. However, we evaluated how Stability Oracle internally views mutating GLY residues via UMAP visualization of the mutation hidden representation and found that Stability Oracle empirically supports your experience by clustering mutations "From" GLY separately for the larger amino acids. Thus, recognizing the unique situation of replacing a GLY residue with a chiral large side-chain (they have been circled in red below but not in the manuscript).

We added Figure 7a to the manuscript to reflect this insight :

Figure 7 a) UMAP visualization of the 128-dim mutation hidden representation for T2837. Left and right panels are colored by the "FromAA" and "ToAA" in a mutation, respectively.

And the following text in page 5, line 185-191, section 2.2:

UMAP visualization of the mutation hidden representation for T2837 from the Structural Amino Acid Embeddings reveals that to the "ToAA" CLS token drives the organization of the latent space and recapitulates known biochemical relationships between the 20 amino acids as illustrated in Figure 7a. We observe that 1) clustering of hydrophobes (LEU, VAL, ILE, MET), aromatics (PHE, TYR, TRP), and short polars (SER, THR and ASP, ASN) (right panel); 2) isolation of the unique amino acids (GLY, CYS, PRO) (right panel); 3) the unique situation of mutating away from GLY and adding a chiral side-chain (left panel). For a residue-specific case study of the 380 mutation types, see Supplementary Figure 7.

Fig. A10: this shows a big cloud, please provide an estimated error

Thank you for highlighting this error on our end. We added the RMSE to the caption:

Supplementary Figure 6: (a) shows Stability Oracle's $\Delta\Delta G$ predictions vs experimental $\Delta\Delta G$ measurements on T2837 + TP (RMSE is 1.53 kcal/mol as displayed in Table 7a above for T2837 Orig + TP). (b) shows the stable prediction subset of (a), where the $\Delta\Delta G$ prediction is < -0.5 kcal/mol.

Regarding evaluation using Pearson, this is another well-known issue. If the authors are convinced that Spearman is more suitable, they should calculate and report Spearman correlation coefficients and not report Pearson yet over again.

While we agree with the reviewer that Spearman is a better suited metric than Pearson, in the main paper we report Pearson primarily to facilitate comparison with the literature for the community. However, all additional metrics, such as Spearman, AUROC, etc, are provided in the supplementary tables, with Spearman being provided in nearly every table. Regarding model development, precision, recall, and AUROC were the primary metrics monitored for improvements.

Category analysis is a somewhat more complicated issue. While it is a valid perspective to say "we'd like to know whether this mutation is stabilising or not", unless neutral and destabilising mutations are grouped into the same category, it would require approaches for 3 categories, which are substantially more complex than the classic ROC curves.

This is a valid point by the reviewer. When we do the stable mutation analysis, we indeed group neutral and destabilizing mutations into 1 class to evaluate its classification performance on stabilizing mutations.

Clustering to avoid data leakage is a good idea, however GraphPart <https://academic.oup.com/nargab/article/5/4/lqad088/7318077?login=false> recommends against MMseq2 as they still detect leakage under its usage.

We thank the reviewer for making us aware of the GraphPart method, upon careful examination of the GraphPart paper, we have concluded that 1) similar to our intuition they used 30% sequence similarity for all but one of their protein dataset, which they did 25% instead because it was a structural prediction task; 2) they devised a more elaborate version of what we did here that can generalize to any training set phenotype. The key distinction here is that we do not conduct K-fold cross-validation splits but rather seed our validation/test set with the established literature $\Delta\Delta G$ test sets (Myoglobin, P53, SSym, S669) to produce T1187. Then, starting from T1187, and similar to GraphPart, we append any homologous data (>30% sequence similarity) to this fold (T1187) to produce at T2837. Finally, and also similar to GraphPart, we used a 30% sequence similarity to T2837 to obtain our cDNA and C2878 training sets. This procedure is depicted in Figure 2.

Additionally, the GraphPart algorithm is not appropriate for our specific datasets because it uses global alignment to accomplish homology partitioning. Here, our training set consists of single domain mini-proteins from the cDNA megascale dataset and our test set T2837 consists of full length multi-domain proteins from pre-established thermostability databases and datasets. This length discrepancy between cDNA and T2837 will result in many false negatives if a global alignment algorithm is used. MMSeqs2 by default uses a local alignment and we ensure at least 50% of the cDNA protein is a part of the alignment.

We looked into why the GraphPart paper reports slightly worse performance with MMSeqs2 compared to other alignment methods. Upon further examination, we determined that this was due to their use of the default sensitivity value of MMSeqs2 (-s 5.7) instead of the maximum sensitivity (s 7.5). Indeed, when we redid our training/test splits with the max MMSeqs2 sensitivity threshold we identified 3 proteins (PDBs: 2M0Y, 2WXC, 6SCW) that leaked into our training sets (sequence similarities: 32.2%, 30.2%, 30.6%), respectively, making the closest sequence overlap between the training test set is 25.4%. To address this minor data leakage, we regenerated our cDNA training set and removed ~3000 mutations from these 3 proteins to produce cDNA117K. Luckily, C2878 was not impacted. We retrained Stability Oracle and Prostata-IFML with cDNA-117K and updated all metrics. We are pleased to see that Stability Oracle and Prostata-IFML performance remained nearly unchanged. Performance comparisons with cDNA120K and cDNA117K on T2837 for Stability Oracle are provided below:

Method	Train Dataset	Pearson	Spearman	RMSE	AUROC	Precision	Recall	Accuracy
Stability Oracle	cDNA120K	0.59	0.62	1.64	0.81	0.53	0.47	0.82
Stability Oracle	cDNA117K	0.59	0.62	1.65	0.81	0.55	0.46	0.82
PROSTATA-IFML	cDNA120K	0.53	0.53	1.75	0.75	0.44	0.46	0.77
PROSTATA-IFML	cDNA117K	0.53	0.52	1.77	0.75	0.43	0.47	0.77

We have made the removal of de novo proteins and these MMSeqs2 configuration more explicit in our methods in page 31, line 662-665, supplementary information B.2:

“First, we removed all *de novo* domains and then filtered all single point mutations on protein scaffolds that have a wildtype structure pdb id provided in the csv. Finally, using structure sequences, the remaining mini-proteins were filtered for 30% sequence overlap with T2837 via MMSeqs2 easy-search command with flags -c 0.3 -s 7.5 --seq-id-mode 1. The closest sequence similarity between cDNA117K and T2837 is 25.4%.”

A lot of the text in the supplementary is a direct copy of the main manuscript text, plus some elaboration - this is not so helpful for readers, please provide additional information only/mainly.

These supplementary texts were meant to elaborate on key issues for interested readers. We have no problem removing them entirely if the reviewers think it is best.

Reviewer #3 (Remarks to the Author):

I co-reviewed this manuscript with one of the reviewers who provided the listed reports as part of the Nature Communications initiative to facilitate training in peer review and appropriate recognition for co-reviewers.

Reviewer #2 (Remarks to the Author):

The authors have addressed most reviewer comments in great detail and thereby I think improved clarity of the paper.

The question of the two input structures for many classical methods still appears misleading to me, as typically the mutant structure is generated by the very same tool that also calculates the ddG. So from a user perspective, only one input structure is needed.

Along similar lines, it should be made clear that StabilityOracle does not produce mutant structures. This can of course be seen as an advantage (in that no wrong model is produced) as much as a drawback. Some users, however, will be looking for models of mutants, e.g. for subsequent design cycles.

In Fig. 6 a red pentagon appears in the plot, however this is missing from the legend, perhaps something was out of sync?

Reviewer #4 (Remarks to the Author):

I agreed with the response of the authors.

I would like to suggest the author update data/script in Git Hub where permission was only given to the reviewers/editors.

There are cases after publication, scripts were not available, or not at a level that could reproduce the results.

Minor suggestions:

line 40, page 2, state-of-the-art => SOTA

line 144-149 repeats what was written in the introduction

line 199, and 200, the values are in 3 decimals, please use 2 as in Figure 5

Reviewer #4 (Remarks on code availability):

Currently <https://github.com/danny305/Stability-Oracle-public> is empty

Revision 2 Rebuttal

Reviewer #2 (Remarks to the Author):

The authors have addressed most reviewer comments in great detail and thereby I think improved clarity of the paper.

The question of the two input structures for many classical methods still appears misleading to me, as typically the mutant structure is generated by the very same tool that also calculates the $\Delta\Delta G$. So from a user perspective, only one input structure is needed.

We thank the reviewer for clarifying what he meant in the previous revision. We agree with the reviewer that some notable classical methods, such as FoldX or Rosetta, will generate mutant structure for the user, making it convenient to the user to only have to provide a single structure. However, these tools still require the generation of the mutant structure, which from the user perspective is computationally intensive and time consuming. Stability Oracle, however, reformulates this step from structure space to embedding space. Thus, completely alleviating the need to compute a mutant structure, irrespective of the tool used, which from the user perspective results in cheaper computation and time required for generating a $\Delta\Delta G$ prediction.

We have further clarified how Stability Oracle uses single structure and amino acid embeddings to eliminate the dependency on a second structure to generate a $\Delta\Delta G$ prediction:

“Previous structure-based stability predictors^{28–30, 57–59} require two structures –either experimental or computational– to explicitly model the wildtype and mutant amino acids. This second (mutant) structure is typically obtained using computational techniques such as AlphaFold or Rosetta. The drawbacks of this approach are 1) computational methods become expensive at inference time (as we describe below) and 2) it is difficult to evaluate the quality of computationally derived mutant structures. In contrast, Stability Oracle does not rely on a second structure. More specifically, structural features from the local chemistry surrounding a particular residue (the masked microenvironment) are extracted from a single initial structure, and a mutation is represented as a pair of “from” and “to” amino acid embeddings vectors. To model the $\Delta\Delta G$ of a specific mutation, the microenvironment of the initial structure is used to contextualize the “from” and “to” amino acid embeddings in the regression head (as illustrated in Figure 1b). This architectural design allows the framework to implicitly learn how “from” and “to” amino acids interact with the local chemistry, rather than relying on a computational structure prediction tool to provide chemical interactions. For a typical 300 amino acid protein, prior work would generate 5700 computational mutant structures (from Rosetta32 or AlphaFold22) in order to predict the $\Delta\Delta G$ of every possible single point mutation during inference. Stability Oracle, on the other hand, requires only one structure to predict the $\Delta\Delta G$ for all 19 amino acid substitutions at every residue (~50 ms/residue). Runtime performance metrics on proteins of varying length are provided in Supplementary Table 1”

Along similar lines, it should be made clear that StabilityOracle does not produce mutant structures. This can of course be seen as an advantage (in that no wrong model is

Revision 2 Rebuttal

produced) as much as a drawback. Some users, however, will be looking for models of mutants, e.g. for subsequent design cycles.

We have further clarified that Stability Oracle does not produce a mutant structure (see response for previous question).

As for subsequent rounds of designs, whatever scaffold a user wishes to make subsequent designs on they can provide an AlphaFold structure (or whatever computational tool of their choice) of that scaffold and generate additional mutational predictions using the AlphaFold structure.

In Fig. 6 a red pentagon appears in the plot, however this is missing from the legend, perhaps something was out of sync?

We apologize for the confusion. Although it appears as a red pentagon, it's a red circle (Stability Oracle) superimposed with a blue pentagon (Prostata-IFML). This is why there are two 0.73 numbers in blue and red next to these markers in the figure.

Reviewer #4 (Remarks to the Author):

I agreed with the response of the authors.

I would like to suggest the author update data/script in Git Hub where permission was only given to the reviewers/editors.

There are cases after publication, scripts were not available, or not at a level that could reproduce the results.

Minor suggestions:

line 40, page 2, state-of-the-art => SOTA

line 144-149 repeats what was written in the introduction

line 199, and 200, the values are in 3 decimals, please use 2 as in Figure 5

We thank the reviewer pointing out these suggestions. We have addressed line 40, 199, and 200.

For line 144-149, we have deleted the duplication and clarified the text to the following:

“First, we built the T2837 test set, which we then used to remove any homologous proteins from the remaining experimental data to produce the C2878 training set. The same procedure was used to construct the cDNA117K training set from the single mutant subset that had experimental structures available of the recently published cDNA-display proteolysis Dataset \#1.”

Reviewer #4 (Remarks on code availability):

Currently <https://github.com/danny305/Stability-Oracle-public> is empty

Revision 2 Rebuttal

We have renamed the github repository to <https://github.com/danny305/StabilityOracle> and have made it publicly available. We have also uploaded it to Zenodo with the following DOI: 10.5281/zenodo.11114641.